# Wireless multi-lateral optofluidic micro-systems for real-time programmable opto-genetics and photopharmacology

Yixin Wu [1,2,3,23], Mingzheng Wu [4,23], Abraham Vázquez-Guardado [2,3,23], Joohee Kim [2,3,5,23], Xin Zhang[4], Raudel Avila[6], Jin-Tae Kim [2,3], Yujun Deng[6,7], Yongjoon Yu[8], Sarah Melzer [9], Yun Bai[1,2,3], Hyoseo Yoon [4], Lingzi Meng[1,2,3], Yi Zhang[10,11], Hexia Guo [1,2,3], Liu Hong[12], Evangelos E. Kanatzidis [2,3,13], Chad R. Haney [14], Emily A. Waters [14], Anthony R. Banks[2,8], Ziying Hu[2,3], Ferrona Lie[8], Leonardo P. Chamorro[12], Bernardo L. Sabatini [9], Yonggang Huang [1,6,15] ✉, Yevgenia Kozorovitskiy [4,16] ✉ & John A. Rogers [1,2,3,8,17,18,19,20,21,22] ✉

In vivo optogenetics and photopharmacology are two techniques for controlling neuronal activity that have immense potential in neuroscience research. Their applications in tether-free groups of animals have been limited in part due to tools availability. Here, we present a wireless, battery-free, programable multilateral optofluidic platform with user-selected modalities for optogenetics, pharmacology and photopharmacology. This system features mechanically compliant microfluidic and electronic interconnects, capabilities for dynamic control over the rates of drug delivery and real-time programmability, simultaneously for up to 256 separate devices in a single cage environment. Our behavioral experiments demonstrate control of motor behaviors in grouped mice through in vivo optogenetics with co-located gene delivery and controlled photolysis of caged glutamate. These optofluidic systems may expand the scope of wireless techniques to study neural processing in animal models.

Technologies to control neuronal activity in vivo are widely employed in neuroscience research for interrogating the function of neural circuits in the nervous system. Neuropharmacology has been used for decades to manipulate neural activity and plasticity via first intrinsic and later genetically modified receptors, providing fundamental insights into neurophysiological processes and pharmacological dynamics[1–4]. In addition, recently established light-based approaches offer greatly enhanced specificity and spatiotemporal precision in modulating targeted neuronal groups or receptors. Optogenetics involves the illumination of genetically modified light-sensitive neurons to activate or inhibit their activity and, thereby, to affect associated physiological processes or behaviors[5,6]. On the other hand,

photopharmacology integrates synthetic chemical photoswitches and light-dependent structural changes to control ligand-receptor interactions, providing high spatiotemporal selectivity without exogenous opsin expression or genetic modifications[7–9]. Tremendous progress has been made in photoswitchable neuroactive compounds, but in vivo photopharmacology remains challenging due to its requirement of controllable delivery of light and drug stimuli to the brain regions of interest[10].

Traditional local pharmacology requires physical connections between a micropump (μ-pump) system and targeted tissues via cannulation and tubing. Conceptually similar schemes exist for optogenetics, with the tubing replaced by optical fibers and the μ-pump

systems by light sources[11]. These tethers and external hardware platforms limit options in multimodal pharmacology and optogenetic studies of freely moving animals, particularly for investigations in complex environments or in interacting groups[12,13]. This latter constraint is significant, because social behavior is essential to individual health and species survival. Understanding the mechanisms by which neural circuit components within and across brain regions synergize to process social information represents a grand challenge in neuroscience[13–16]. Advanced platforms that offer bidirectional control over light and drug delivery, independently across animals in collective groups, with more than a single wavelength or a single pharmacological compound, may represent promising technologies in this context[17–22]. For example, prior studies indicate that bidirectional modulation can guide synaptic plasticity to strengthen and weaken alcohol-seeking or cocaine craving behaviors[23,24]. Other work demonstrates similar modes of modulation over fear, anxiety, and social interactions that range from grooming to fighting[25–27].

Wireless neurotechnologies overcome many limitations of traditional tethered systems[28–33]. Initial work in wireless optofluidics led to demonstrations of battery-powered devices with thermal-mechanical μ-pump in head-stage mounting configurations[34]. Subsequent advances served as the foundations for lightweight, battery-free and fully implantable optofluidic designs[35,36], although without the ability for flow modulation, real-time control, or independent addressability of multiple optofluidic devices in a single operating environment. These constraints limit the applications of wireless neuropharmacology in behavior studies.

This paper introduces a wireless, battery-free, miniaturized multi-channel optofluidic platform for optogenetics and photopharmacology, designed specifically for use with small, freely moving animals in isolation or interacting groups in large enclosures. Key features of this system overcome limitations of alternative technologies (Supplementary Table 1) through the use of: (1) customizable, multilateral probes with soft μ-fluidic/electronic interconnections that support versatility in positioning across the nervous system, enabling either synchronized or desynchronized patterns of modulation in the same or different regions; (2) passive μ-fluidic check valves that allow precise control over flow dynamics; (3) designs that support capabilities for real-time, independent multichannel control of all relevant parameters in fluidic delivery and optical illumination, as well as individual addressability over multiple devices (up to 256) within a single experimental field via near-field communication (NFC); (4) engineering choices that leverage low-cost commercial components (Supplementary Table 2), scalable manufacturing and assembly procedures, user-friendly open-source graphical user interfaces (GUIs) and that enable simple implantation methods to facilitate broad use by the neuroscience community. Demonstrations of these multilateral optofluidic devices for wireless control of behaviors in individual and grouped mice through optogenetic and photopharmacological approaches illustrate advanced modes of function that are unavailable in alternative platforms and are essential for modern experiments in neuroscience research.

## Results

### Design of a wireless, battery-free, multi-lateral, multi-modal optofluidic platform

The device platform supports wireless programmable multi-channel fluidic delivery and light stimulation via a battery-free, low-power, electronic module (10 mm × 13 mm, thickness: 1.5 mm). Key components are a set of two lightweight, miniature μ-pumps, and a pair of mechanically compliant μ-fluidic probes (cross-section: 250 μm × 150 μm; polydimethylsiloxane (PDMS)) (Fig. 1a). These probes can incorporate microscale inorganic light-emitting diodes (μ-ILEDs, 270 μm × 220 μm × 50 μm) attached at their tip ends through serpentine electrical interconnections to produce colocalized brain tissue

illumination (Probe total thickness: ~350 μm; Distance between μ-fluidic and center of μ-ILED: ~120 μm, Supplementary Fig. 1), ensuring the spatial proximity required for viral transduction based optogenetics and photolysis of locally infused compounds. This system is sufficiently thin and lightweight (0.1514 grams) (Supplementary Fig. 1) to enable wireless, battery-free and programmable operation for free-moving animal studies, with no measurable effect on locomotor activity[36]. This technology leverages commercially available electronic components and established, low-cost manufacturing processes, enhancing the potential for broad distribution to the neuroscience community.

Multi-channel modules offer various possibilities in experimental protocols that rely on programmable pharmacology and/or optogenetics. Up to 4 channels for either optical stimulation or fluidic delivery can be integrated into either unilateral or bilateral probes for different purposes. The bilateral module has two probes that can be implanted into separate hemispheres for either synchronized or desynchronized patterns of optical and/or pharmacological modulation in the same or different brain regions. Each probe has one μ-fluidic channel for drug delivery and one μ-ILED for single-wavelength optical stimulation (Fig. 1b–d, Supplementary Fig. 1). The unilateral module includes two μ-fluidic channels and two μ-ILEDs with the same or different wavelengths, all on a single probe (Fig. 1e–g and Supplementary Fig. 1). This unilateral device can support excitation and/or inhibition using (1) optogenetics at two wavelengths; (2) pharmacology with two drugs; (3) optogenetic and photopharmacology simultaneously, at a single location. The flexible serpentine interconnections and elastomeric μ-fluidic channels that link the probes to the circuit platform and the micro-reservoirs (μ-reservoir)/ μ-pump chambers, respectively, pass through an open area of the device to allow straightforward insertion into a brain region of interest (Supplementary Fig. 2). The lengths and widths of these probes can be selected over a wide range of dimensions to meet requirements of implantation in different brain regions (Supplementary Fig. 2). After insertion of the probes, the main body of the device, defined by the electronic module and μ-reservoirs/μ-pump chambers, rests on the top of the skull, secured by dental cement. Further details of the implantation process are in the Methods section. A photograph of a mouse with an implanted bilateral device is shown in Fig. 1h. The computed tomography (CT) image in Fig. 1i shows a bilateral device implanted in a mouse, with the probes inserted in both hemispheres.

### Real-time programmable operation of a multi-lateral, multi-modal optofluidic platform for advanced neural control

Capabilities in real-time, programmable control create unique options in studies of freely moving animals that exploit optogenetics, pharmacology and photopharmacology[18,35,36]. Here, an intuitive GUI that operates on a personal computer or tablet exploits standard NFC communication protocols (ISO 15693) for reading and writing information as control commands to guide the logic flow of the firmware in the microcontroller (μC) (Fig. 1j and Supplementary Fig. 3). Previous demonstrations utilized devices with pre-set parameters, limiting the number of available protocols[35]. In contrast, this device offers a multimodal platform with a diverse set of operation and reconfiguration capabilities accessible by the user via the GUI in real-time and on an event basis. Most importantly, the independent operation of four channels that connect to two μ-ILEDs and two μ-pumps in unilateral or bilateral configurations serves as the basis for optogenetics, pharmacology or photopharmacology modalities, of relevance for neuroscience research.

The versatility of this multimodal platform permits programmable operation in both bilateral and unilateral optofluidic control. Upon sending the desired parameters via NFC, the μC activates the electrochemical μ-pumps for fluidic delivery or the μ-ILEDs for optical stimulation either individually or simultaneously, each at a desired

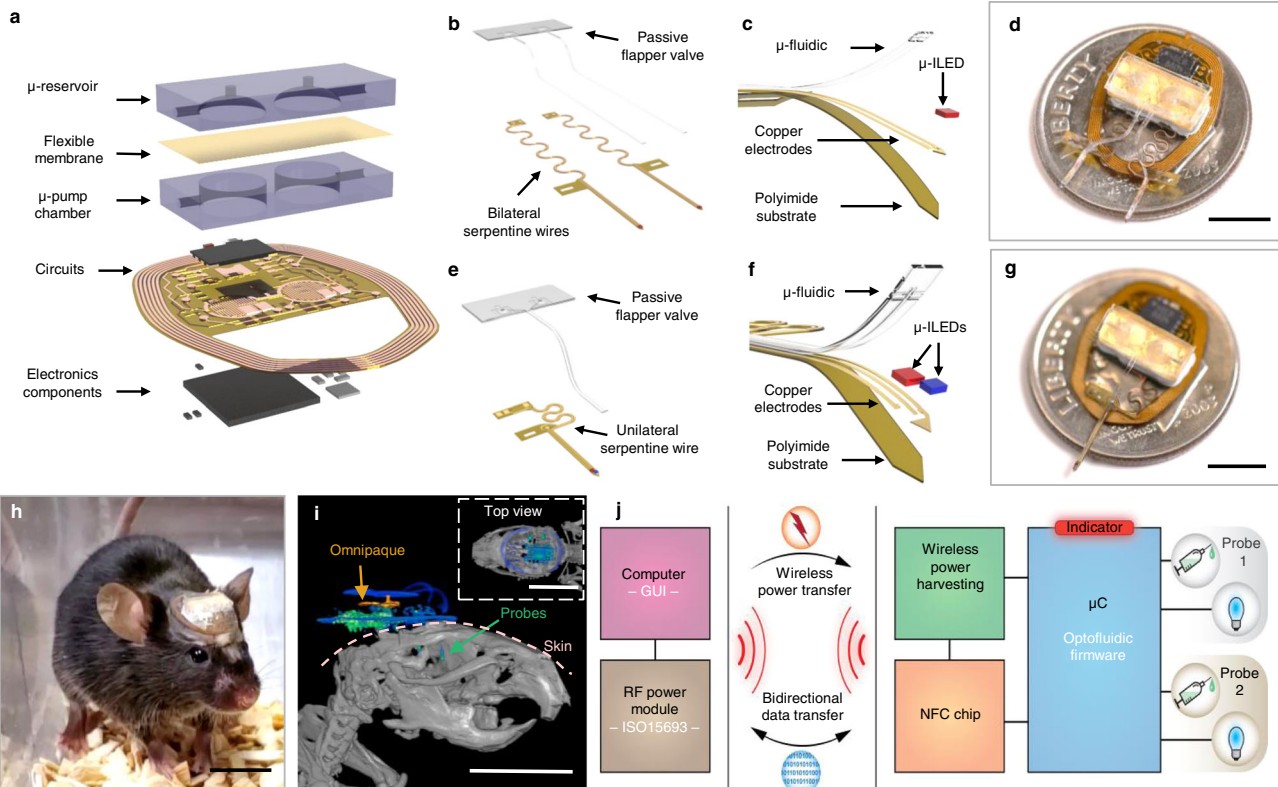

**Fig. 1 | Wireless, battery-free multi-lateral microsystems for programmable pharmacology and optogenetics. a** Diagram illustration of the system. **b, c** and **e, f** Magnified views of the bilateral and unilateral neural probes with optical (μ-ILED) and fluidic delivery. **d** and **g** Photographic images of the front side of both bilateral and unilateral bi-channel devices. Scale bar: 0.5 cm. **h** Photos of the optofluidic device mounted on a mouse. Scale bar: 0.5 cm. **i** CT images of the side view and top view of the implanted optofluidic device on a mouse brain with iodinated contrast (Omnipaque) to demonstrate the location of the μ-reservoirs. Scale bar: 1 cm. **j** Schematic diagram of the electronics design for programmable and independent control over different channels.

frequency and duty cycle. In single-channel mode, the corresponding general purpose input output (GPIO) pin of the μC applies a train of voltage pulses at the programmed frequency and duty cycle to activate the μ-ILEDs or the μ-pumps. In dual-channel mode, these pulses occur out-of-phase to manage power consumption and ensure sufficient supply to each of the two arbitrarily selected channels (Fig. 2a). A red indicator LED located on the back of the device serves as a visual signal that blinks with different frequencies to indicate the current mode of operation, facilitating concurrent behavioral analyses.

Various modes of operation of the optofluidic platform and its applications in optogenetics and neuropharmacology appear in Fig. 2b–i. The basic features include independent channel operation for activation/inactivation and the control of μ-ILED and μ-pump and their parameters (frequency and duty cycle). Simple illustrations and experiments using colored aqueous solutions and transparent brain tissue phantoms (0.6% agarose gel) summarized in Fig. 2b–d demonstrate these functions in bilateral optofluidic devices. The examples include unilateral dual-μ-ILED operation (Fig. 2e, f), unilateral dual-μ-pump operation (Fig. 2g, h) and unilateral/bilateral optofluidic operation (Fig. 2i, j). The unilateral dual-μ-ILED operation, for instance, allows the simultaneous control of two spatially co-expressed opsins through two μ-ILEDs of different wavelengths (Fig. 2e, f), whereas the unilateral dual-μ-pump operation enables the delivery of two different neuroactive compounds at the same location (Fig. 2g, h). These two modalities can facilitate bidirectional studies using optogenetics and neuropharmacology by applying programmable excitatory and inhibitory stimuli. Finally, the combined optofluidic operation provides optical and pharmacological multimodal control at a single location (Fig. 2i). A demonstration that involves fluorophore infusion in transparent phantom brain tissues (0.6% agarose gel) followed by blue light

stimulation to induce fluorescent emission shown in Fig. 2j, illustrates operation relevant to virus infusion followed by optical control of genetically modified neurons for optogenetics, or drug delivery followed by light-induced uncaging for photopharmacology.

Finally, the unique device identification (UDID) number encoded in each NFC chip enables individual device addressability, as the basis for control over multiple devices within the same field, all through a single GUI (Supplementary Fig. 4). The RF module allows the discovery of up to 256 devices within the RF field during the inventory command. This feature is fundamental to studies of social interactions among groups of animals with optical and/or pharmacological neuromodulation. A simple illustration that demonstrates this capability uses three devices implanted into three different mice, each with different color indicators for visual confirmation. The GUI provides real-time control over each device, selectively and independently (Fig. 2k, l, Supplementary Video 1).

## Design of an electrochemical μ-pump and μ-fluidic passive flapper valve for dynamic flow control

The fluidic system includes a low power (<1 mW) (Supplementary Fig. 5) miniaturized refillable electrochemical μ-pump, a thin passive flapper valve for flow control and a narrow, thin elastomeric μ-fluidic probe for fluid delivery[35,36]. The electrolysis approach is attractive for its low power consumption, minimal heat generation (<0.2 °C), large driving force and simple construction (Supplementary Fig. 6). The specific layout described here includes a set of interdigitated electrodes (Au/Cu, 200 nm/18 μm, ~50 μm space), a μ-pump chamber (cylinder shape.; height 1 mm; diameter 2.45 mm) filled with an aqueous electrolyte solution (potassium hydroxide (KOH); 50 mM), a gold-coated flexible polystyrene-block-polyisoprene-block-polystyrene (SIS) membrane, and a dome-shaped μ-reservoir (volume: 500 nL)

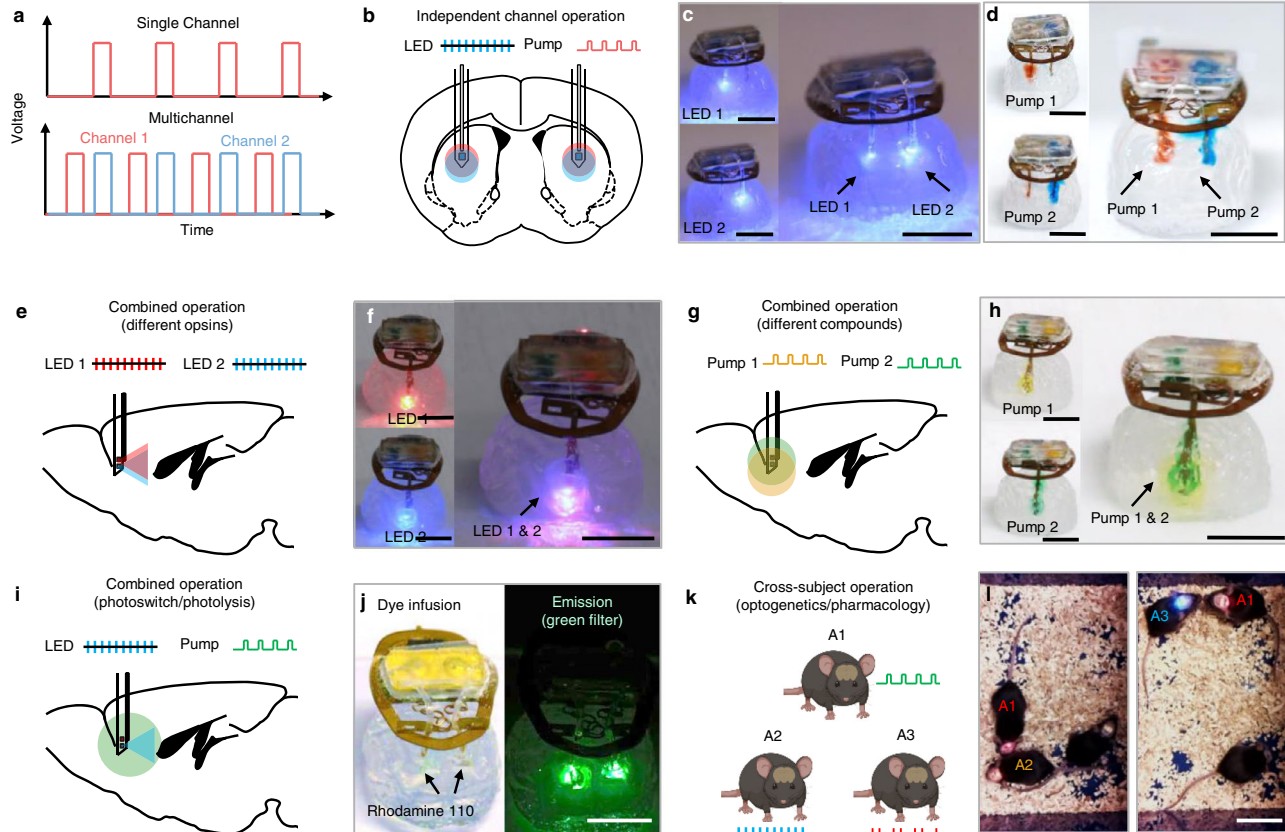

**Fig. 2 | NFC-enabled dynamically programmable and selective operations for both bilateral and unilateral bi-channel devices. a** Waveforms associated with the single mode and out of phase dual mode of operation. **b** Illustration of independent channel operation of a single μ-ILED or μ-pump or dual μ-ILEDs or μ-pumps in a bilateral optofluidic device. **c, d** Photographs that demonstrate bilateral optofluidic devices operated in single mode and dual mode for both optical stimulation (4 Hz, 50% duty cycle) and fluidic delivery (4 Hz, 10% duty cycle). Scale bar: 0.5 cm. **e** Illustration of combined operation of two μ-ILEDs with different wavelengths for different opsins stimulation at a single location. **f** Photograph that demonstrates unilateral bi-channel optofluidic devices with single/dual two-wavelength optical stimulation (4 Hz, 100% duty cycle).Scale bar: 0.5 cm. **g** Illustration of combined operation of two individual μ-pumps for various compounds infusions at a single

location. **h** Photograph that demonstrates a unilateral bi-channel optofluidic with single/dual modes of liquid infusion in hydrogel phantom brain tissue (4 Hz, 100% duty cycle). Scale bar: 0.5 cm. **i** Illustration of combined operation of an μ-ILED and μ-pump for photoswitching and photolysis studies. **j** Photographs of fluorophore infusion through μ-pump actuation (left, 4 Hz, 50% duty cycle) and green fluorescent emission of the fluorophore after stimulation by blue μ-ILEDs (right, 4 Hz, 10% duty cycle). Scale bar: 0.5 cm. **k** Illustration of interbrain operation for individual control of each animal for optogenetics or pharmacology within a group of animals. **l** Photos of individual control over multiple devices in the field for interaction studies. Mouse 1 (red indicator) and mouse 2 (orange indicator) are activated in the left photo, while mouse 1 and mouse 3 (blue indicator) are activated in the right photo. Scale bar: 5 cm.

filled with drug (Supplementary Fig. 7). At room temperature, a voltage applied to the interdigitated electrodes initiates the reaction $2H_2O$ (liquid) → $O_2$ (gas) + $H_2$ (gas). The resulting gas pressure deforms the flexible membrane, thereby driving the flow of liquid from the μ-reservoir through the passive flapper valve and down the lengths of the μ-fluidic channels to the target area of the brain (Fig. 3a). The gold coating on the electrodes prevents corrosion from the electrolyte and the electrolysis process (Supplementary Fig. 8). The materials for the μ-pump chamber and μ-reservoir have low gas and water vapor permeability and absorption, as well as chemical inertness to substances relevant to the applications considered here. Refilling ports on the sides of both the μ-pump chamber and μ-reservoir can be sealed with silicone elastomer (Kwik-Sil, World Precision Instruments) and opened/resealed for multiple cycles of use (Supplementary Fig. 8).

The flexible membrane that separates the μ-pump chamber from the μ-reservoirs includes a multilayer coating of $Ti/Au/Ti/SiO_2$, to reduce the rate of gas permeation (Fig. 3b). Without this gold coating, the film can quickly relax back to its original geometry due to transmembrane permeation of trapped gas and to recombination of oxygen and hydrogen molecules in the electrolysis reaction[37,38]. The titanium layers promote adhesion between the gold and polymer substrate as well as the gold and $SiO_2$ layer. The $SiO_2$ renders a hydrophilic surface

to avoid bubble formation in the drug reservoirs. This relaxation can potentially draw drugs along with biofluids near the tip end of the probe back along the lengths of the microchannels (Supplementary Fig. 9). A thin, multi-layer, passive μ-fluidic check valve based on a flapper type design minimizes this effect. This valve includes a bottom layer with an inlet and a circular step structure, an elastic flexible membrane (25 μm, PDMS) with a circular hole in the middle of the circular step, and a top circular layer that aligns with the bottom structure and connects to the outlet (Fig. 3c). Liquid pressure in the forward direction causes deflection, allowing the fluid to flow through the hole. The film is thin and easily deformed, but with sufficient stiffness to return to the closed state in the absence of pressure. In the presence of pressure from the outlet, the valve remains in this closed state to prevent flow in the backward direction (Supplementary Fig. 9). The steps for fabricating the passive flapper valve are in Supplementary Fig. 10. Theoretical modeling defines optimized choices of dimensions and design layouts (Supplementary Fig. 11), subject to practical limits in the fabrication process. Demonstration experiments highlighted in Fig. 3d, show that red dyed water pumped from the inlet easily passes through the valve, while blue dyed water pumped from the outlet cannot pass. Quantitative measurements indicate that the forward pressure needed to initiate flow is around 7.2 kPa. Backward

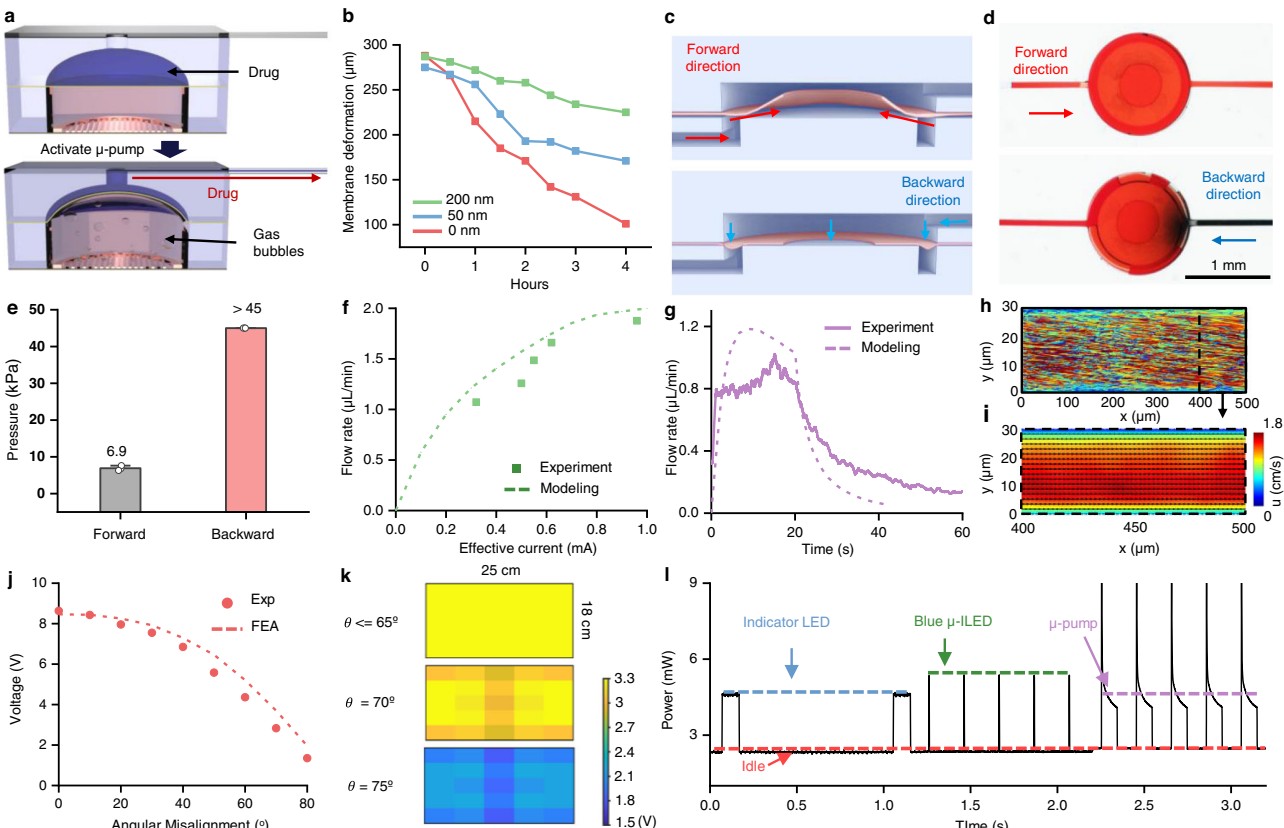

**Fig. 3 | Fluidic and electronic characteristics of multi-model wireless opto-fluidic system. a** Schematic diagram of the electrochemical μ-pump actuated by electrolysis reaction. **b** Relaxation of the film with different thicknesses of gold coatings (0/50/200 nm) following actuation after 2 min of μ-pump activation. **c**, **d** Schematic illustrations and photographs of the passive flapper valve with flow from forward and backward directions. In the photos, the red aqueous solution flows through the valve from the forward direction with low resistance, while even under high pumping pressure, the blue aqueous solution is unable to pass. The blue color is from the blue dye diffusion in the top layer. **e** Burst pressure required for flow through the passive flapper valve from the forward and backward directions. $n = 3$. Data are presented as mean values ± SEM. Backward direction pressure data from three independent samples exceed the upper limit of the instrument used for measurement. **f** Flow rate as a function of effective current. **g** Temporal profile of the flow rate when operated at 250 ms frequency and 10% duty cycle. **h** Flow trajectories at 10 s < $t$ < 20 s of (**g**); Color denotes the streamwise velocity, $u$. **i** Averaged Eulerian velocity contour of (**h**) at 400 um < $x$ < 500 μm with super-imposed vector field. **j** Angular misalignment effect on power transfer efficiency at the center of the field at 3.81 cm height. **k** Power distribution of the field at height 3.81 cm with different angular misalignment (0–75°). **l** Power consumption of four states of the device in the cage with 8W RF power applied on the transmission antenna. Four states include: Low power μC idle state, indicator LED with 1 Hz frequency and 10% duty cycle, blue μ-ILED with 5 Hz frequency and 1% duty cycle and μ-pump activation with 5 Hz frequency and 45% duty cycle. Source data are provided as a Source Data file.

flow can be stopped completely even for pressures >45 kPa, well above the values needed for practical applications (Fig. 3e). Based on simulation results (Supplementary Fig. 12) and experimental observation, the μ-pump can easily generate pressures (up to ~30 kPa) above the threshold for initiating forward flow.

### Fluidic and electronic characteristics for multi-modal wireless optofluidic operation

For many applications, the ability to control the flow rate is important. For instance, the flow rate must be sufficient to initiate desired responses on a triggered basis, but it must remain below levels that could induce damage to adjacent healthy tissue and avoid backflow along the insertion tract to eliminate the off-target effect[39] (Supplementary Fig. 13). Selection of flow rate and μ-reservoir size must consider rates of diffusion and metabolism of the drug, along with its target distribution volume[40]. A precise understanding of the flow characteristics of the system is, therefore, important.

Experimental characterization of the flow rate of the device at various duty cycles (Supplementary Fig. 14) relies on high-speed micro-Particle Tracking Velocimetry (μ-PTV). PTV tracks the flow particles in a Lagrangain frame of ref. 41 as devices under pulse width modulations with different duty cycles deliver aqueous solutions mixed with fluorescent particles (pink, diameter: 0.86 μm, Sphero) as tracers through μ-fluidic channels (width: 30 μm; height: 30 μm; length (~7 mm)) (Fig. 3f). As expected, representative trajectories at 10% duty cycle exhibit characteristics of a pressure-driven closed-channel flow (Fig. 3g). A total momentum flux deficit from the boundary layer represents a major source of the μ-fluidic resistance, accounted for in the simulation. Further investigations use Lagrangian flow trajectories transformed into the Eulerian velocity field (Fig. 3h, i). After 10–20 s, the flow is fully developed, as defined by velocity profiles that do not change along the flow direction. An analytical approach that exploits the rate form of the ideal gas law (Eq. 1)[40] and accounts for the geo-metrical, flexural, and fluidic parameters can capture behaviors for different operational scenarios (e.g., duty cycle/effective current). Based on the stress-strain relationship of the SIS flexible membrane (Supplementary Fig. 12), the temporal profile of the flow rate can be determined from Eq. 4. Both the maximum flow rate (peak value of the temporal profile) vs effective current curve and the temporal profile of the flow rate (Fig. 3f, g) obtained in this way reproduce experimental measurements of device behavior for different operating conditions (~9% average error). The measurement uncertainties are likely due to insufficient or variable numbers of tracking particles and/or non-uniform distributions of these particles due to aggregation or

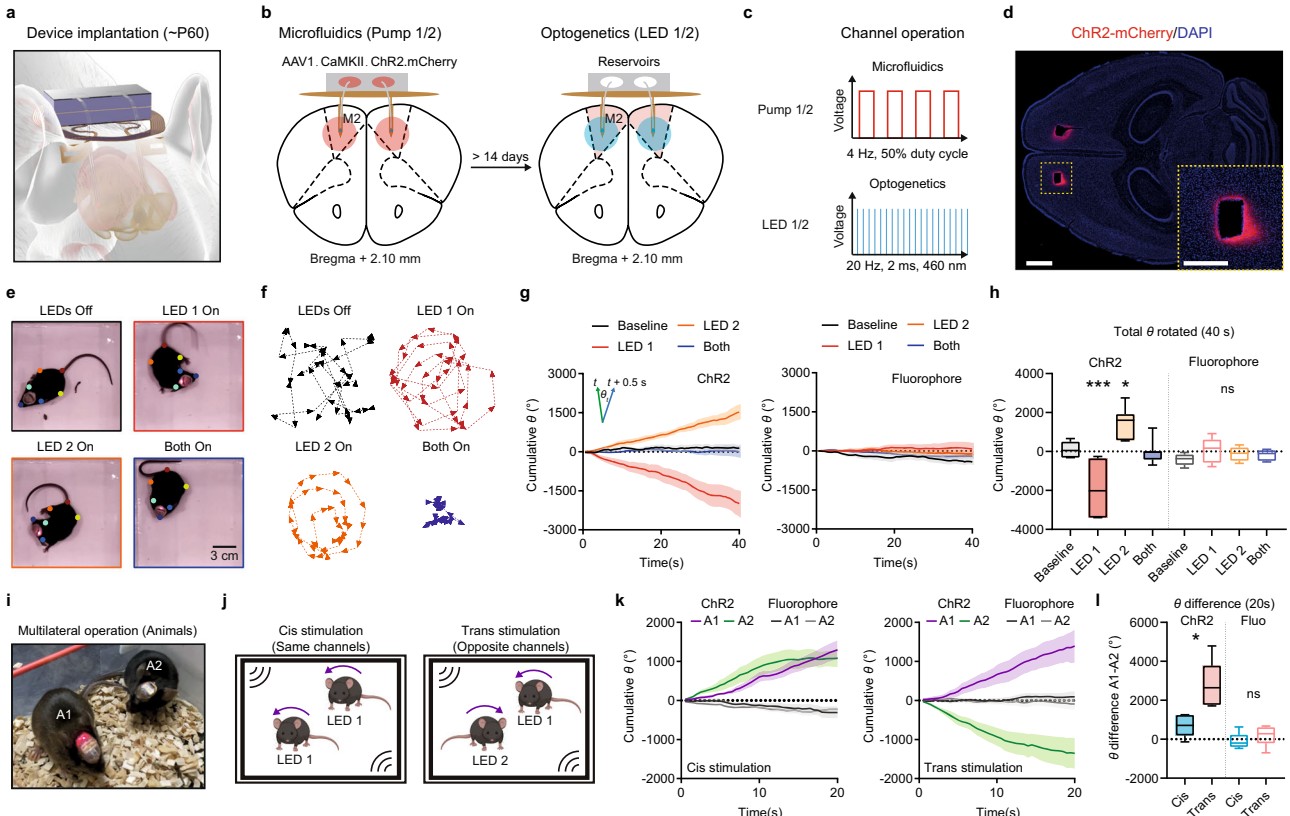

**Fig. 4 | Optogenetic control of motor behavior of individual and grouped mice using multilateral optofluidics device. a** Schematic illustration of device implantation on the skull of a mouse model. **b** Schematic illustration of virus transduction and experimental timeline. **c** Parameters of microfluidic delivery and optogenetic stimulation for individual channels. **d** Example image of ChR2-mCherry expression in M2 after virus transduction using microfluidic channels. Scale bar: 1 mm and 500 μm. **e** Example frames of one ChR2-expressing mouse during controlled optogenetic stimulation using different channel operations. **f** Example traces of one ChR2-expressing mouse in a 20-second-long episode with different channel operations. **g** Left, summary data showing the cumulative rotation degrees in mice expressing ChR2 during a 40-second-long episode. Right, same as left, but for mice expressing control fluorophore. Inset, calculation of rotation degree (θ) for each frame (0.5 s bin). *n* = 7 animals/group. Data are presented as mean values ± SEM. **h** Summary data showing the total degrees rotated in each experimental condition. One-way ANOVA, ChR2, *p* < 0.0001, F (3, 24) = 20.16; Fluorophore, *p* = 0.1858, F (3, 24) = 1.739. Sidak's multiple comparison, ChR2 (vs Baseline), LED 1, *p* = 0.0003, LED 2, *p* = 0.0173, Both, *p* = 0.9855; Fluorophore (vs Baseline), LED 1, *p* = 0.1014, LED 2, *p* = 0.4071, Both, *p* = 0.6926. *n* = 7 animals/ group. Bounds of box show quartiles and median, whiskers show minima and maxima. **i** Example image of multilateral device operation in a pair of mice. **j** Schematic illustration of the experimental design to independently control motor behaviors in paired mice. **k** Left, summary data showing the cumulative rotation degrees in mice expressing ChR2 and fluorophore during a 20-second-long Cis operation. Right, same as left, but for Trans operation. *n* = 5 animal pairs in ChR2 group and six animal pairs in fluorophore group. Data are presented as mean values ± SEM. **l** Summary data showing the difference of degrees rotated between the two mice in each experimental condition. Unpaired two-tailed *t*-test, Cis vs Trans, ChR2, *p* = 0.0103, Fluorophore, *p* = 0.3304. *n* = 5 animal pairs in ChR2 group and six animal pairs in fluorophore group. Bounds of box show quartiles and median, whiskers show minima and maxima. *\*p* < 0.05, *\*\*\*p* < 0.001. Source data are provided as a Source Data file.

accumulation at the channel boundaries. The results of modeling confirm the capability to control the delivered volume through the duty cycle and delivery time (Supplementary Fig. 14h, i). Using a simple GUI, the duty cycles, delivery times, delivery cycles and interval times can be easily configured, thereby allowing multi-dose animal experiments. Notably, the fluidic characteristics may differ in the brain because of uptake and active clearance of the infused agents, as well as the inhomogeneous and highly anisotropic nature of brain tissue compared to gel. Despite these differences, previous literature demonstrates that the agarose gel model is useful as a simplified version of the brain for preliminary estimates of flow[42]. Further details are in the Methods section.

Stability in wireless power delivery is essential for reliable electrical performance and consistent operation. The setup reported here utilizes an optimized double-transmission loop antenna that yields a uniform magnetic field distribution within the experimental volume to provide constant power to the devices, consistent with numerical simulations shown in Fig. 3j and Supplementary Fig. 15. The NFC chip utilizes ~0.1 mW during stand-by mode, which corresponds to the standard mode of operation unless a read/write command is performed. The μC, on the other hand, consumes ~2.22 mW during its operation. Power consumption can be further reduced when implementing additional power-saving modes such as those reported in other similar systems (0.04 mW average and 0.4 mW peak powers)[36]. A double-sided coil (12 turns) receiving antenna maximizes power harvesting efficiency and yields a high-quality factor (*q* = 24.51) in a size-constrained configuration set by the dimensions of the skulls of mice. This antenna design harvests up to 54.26 mW of power when placed at the center of the experimental enclosure, the location with the smallest magnetic field density. The electrical power harvested from this double-sided coil antenna with large q factor allows for significantly enhanced stability in near-field communication compared to that possible with single-sided coils (6 turns) of the same surface area, as shown in Supplementary Fig. 16. The effect of angular misalignment between the receiver coil and the direction of the magnetic field generated by the transmission antenna is also important. Numerical and experimental results of a device located in the middle of the enclosure at a height of 3.8 cm from the base indicate a stable voltage

output for angles up to 70°. Other advanced antenna designs can almost completely eliminate these angular effects[43]. The spatial uniformity can be visualized by mapping the voltage output at different angle orientations (0°–75°) and different heights (2.54 cm and 3.81 cm) (Fig. 3k and Supplementary Fig. 17).

Power consumption characteristics, measured during four different states of operation within the field, appear in Fig. 3l. These modalities include (1) idle, corresponding to inactive μ-ILEDs and μ-pumps, (2) operation of the red indicator LED, (3) operation of the blue μ-ILED and (4) operation of the μ-pump. During idle mode, the μC and NFC chip consume 2.32 mW, as the baseline in the power consumption graph. The peak power consumption of the red indicator LED (frequency: 1 Hz, duty cycle: 10%) is 2.29 mW, the blue μ-ILED (frequency: 5 Hz, duty cycle: 1%) is 3.05 mW, and the μ-pump at a high duty cycle (frequency: 5 Hz, duty cycle: 45%) is 2.3 mW (Supplementary Fig. 18). In all cases, the average power consumption decreases in an expected manner with duty cycle.

## Biocompatibility and positioning accuracy of the flexible optofluidic probes

Tests of the biocompatibility of the probes involve comparison of reactive gliosis between μ-fluidic probes and conventional metal cannula implants using immunohistochemical markers of glial cells. Mouse brains were extracted 7 days after implantation and examined with immunostaining. Mice implanted with μ-fluidic probes or 500 μm diameter metal cannulas show similar astrocytic (GFAP) activation (Supplementary Fig. 19). Notably, significantly lower activation of microglial cells (IBA1) and smaller damage areas in horizontal brain slices are observed for subjects implanted with μ-fluidic probes (Supplementary Fig. 19). Lower microglial activation may potentially result from a decreased amount of brain tissue damage with the probes[44]. These data show that μ-fluidic probes decrease local disruption to the targeted brain regions compared to the conventional cannulation.

To estimate positional accuracy for the flexible optofluidic probes, we targeted the same brain region in the two hemispheres using (1) two separately implantable probes (Flexible design), and (2) an integrated probe with two tips (Fixed design) (Supplementary Fig. 20). After implantation, we fixed the brain tissue and performed histological assessments of the displacement between two probe tips. We set one probe location as the reference (X0, Y0), and measured the distance from the center point in the other hemisphere (XY in absolute value) of the other probe location. Our result shows that the displacement (mean ± SEM) in the X-axis (medial-lateral axis) is 42.05 ± 10.42 μm in the fixed design and 153.6 ± 48.24 μm in the flexible design, while the displacement in the Y-axis (anterior-posterior axis) is 108.1 ± 18.26 μm in the fixed design and 84.14 ± 26.14 μm in the flexible design (Supplementary Fig. 20). Therefore, we introduced the fixed probe option to reduce the surgical complexity based on user preference, although the individually implantable probes provide greater flexibility to target different combinations of brain regions of interest.

## Optogenetic and photopharmacology control of motor behavior in individual and grouped mice

To demonstrate the capability of these wireless multilateral optofluidic devices, we aimed to control motor behavior in freely moving mice through independent multi-channel operations. We targeted the secondary motor cortex (M2) through a single standard implantation procedure (Fig. 4a and Supplementary Fig. 21). The device configuration used in this demonstration consisted of two bilateral probes each with optogenetics and drug delivery capability: probe one/two with LED 1/LED 2 (optogenetics) and Pump 1/Pump 2 (drug delivery), which are implanted into the right-M2/left-M2, respectively. Upon activation, M2 drives robust motor behaviors that can be reliably observed and quantified using machine-learning based pose estimation algorithms (Deeplabcut[45])[46].

AAV1.CaMKII.ChR2.mCherry virus solution was loaded into the two μ-reservoirs prior to device implantation. The virus was subsequently delivered into right-M2 (Pump 1 of the device) and left-M2 (Pump 2 of the device) through μ-fluidic operation after mice recovered from anesthesia. Through a single implantation procedure, the optofluidic probe ensures precisely localized viral expression to the μ-ILEDs and μ-fluidic portals. Two weeks after device implantation and virus transduction, we tested the motor behavior of mice using optogenetic stimulation (Fig. 4b, c). The expression of ChR2.mCherry was validated by histology after behavioral assays (Fig. 4d). As expected, we induced substantial rotational behavior with wireless unilateral optogenetic stimulation in M2. The direction of rotational behavior was controlled in a real-time programmable manner by altering the activated channel (LED 1, right-M2, left-turning behavior; LED 2, left-M2, right-turning behavior). When both channels were activated simultaneously, mice did not increase rotational behavior, presumably because of the cancellation of left and right turning motor commands (Fig. 4e–g, Supplementary Video 2). The implementation of the same stimulation protocol on the control subjects expressing a fluorophore did not elicit significant rotational behavior, suggesting that rotation did not result from opsin-independent effects induced by light stimulation (Fig. 4g, h).

Wireless neuroelectronic devices eliminate the physical constraints induced by externally tethered cables, creating a unique opportunity for neuroscience researchers to study behaviors in groups of small animals. The integrated NFC chip of the optofluidic devices allows experimenters to address different devices independently, providing the capability to induce synergistic or antagonistic behavior among animals. To demonstrate multilateral operation in groups, we used differently combined channel operation regimes to induce distinct rotational patterns in paired mice. During Cis stimulation, the same channels in paired mice in the same arena were activated, while the opposite channels were activated during Trans stimulation (Fig. 4i, j). Synergistic rotation was induced in paired mice during Cis operation, and antagonistic rotation was induced during Trans operation. Again, no significant motor behaviors were induced in the fluorophore expressing control group (Fig. 4k, l).

The development of caged compounds and light-based uncaging methods allows researchers to mimic the release of neurotransmitters and neuromodulators with spatiotemporal precision[7,47–54]. The emergence of photopharmacology, relying on the photolysis of caged compounds with light, opens new approaches to dissecting the function of neuroactive molecules and developing targeted therapeutics[10,55]. The optofluidic device designed in this study enables in vivo photopharmacology by combining independent light stimulation and drug delivery, a capability that has not been achieved by any other previously reported wireless neurotechnologies. To demonstrate this functionality, we used Rubi-glutamate[56], a caged glutamate compound that undergoes photolysis with blue light stimulation (460 nm), to modulate motor behavior in mice (Fig. 5a). Ex vivo voltage clamp recording of an M2 pyramidal neuron shows that Rubi-glutamate reliably evokes excitatory postsynaptic currents (EPSCs) upon light stimulation (460 nm, 20 ms). Uncaging-evoked EPSCs (uEPSCs) were completely blocked by the co-application of glutamate receptor antagonists NBQX (AMPA-R) and CPP (NMDA-R), confirming their glutamatergic identity (Fig. 5b). We tested whether the optofluidic device can provide sufficient power to photolyze Rubi-glutamate in vitro using nuclear magnetic resonance (NMR) spectroscopy. Our results showed that the concentration of glutamate increased substantially after light stimulation for 0–30 min (Fig. 5c and Supplementary Fig. 22). Next, we performed in vivo photopharmacology experiments targeting right M2 with μ-fluidic and controlled light stimulation. Rubi-glutamate was diffused by the μ-fluidic probe and blue light was subsequently delivered at 20 Hz (20 ms width, 460 nm) (Fig. 5d). Mice showed a significant preference for left-turning rotation versus right-turning during light stimulation

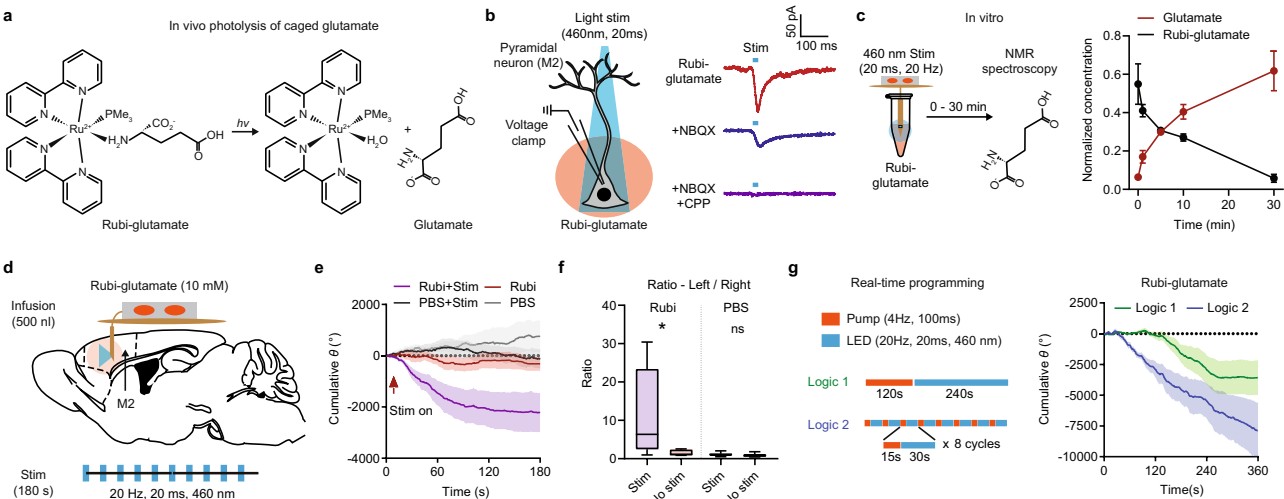

**Fig. 5 | Optopharmacological control of motor behavior using multilateral optofluidics device. a** Chemical formula showing the photolysis of Rubi-glutamate. **b** Left, schematic showing voltage clamp recording of a pyramidal neuron in M2 upon photolysis of Rubi-glutamate using 460 nm light stimulation. Right, example of EPSCs evoked by photolysis of Rubi-glutamate (50 μM) Holding potential, −70 mV. NBQX and CPP, 10 μM. Light stimulation, 460 nm, 20 ms pulse width. **c** Left, schematic showing in vitro photolysis of Rubi-glutamate using optofluidic devices. Concentration of Rubi-glutamate and glutamate were measured by Nuclear magnetic resonance (NMR) spectroscopy following light stimulation (0–30 min, 460 nm, 20 ms pulse width, 20 Hz frequency). Right, normalized concentration of Rubi-glutamate and glutamate after different duration of photolysis. $n = 3$ independent experiments. **d** Schematic showing the experimental design of optopharmacology using Rubi-glutamate in vivo. **e** Summary data

showing the cumulative rotation degrees in mice infused with Rubi-glutamate or PBS, with or without light stimulation. **f** Summary data showing the ratio of the degrees rotated in left direction over right direction. Two-sided Mann–Whitney test, Stim vs No stim, Rubi-glutamate, $p = 0.0303$, PBS, $p = 0.3829$. $n = 5$ animals in Rubi+stim group, six animals in Rubi only group, and seven animals in PBS groups. Bounds of box show quartiles and median, whiskers show minima and maxima. **g** Left, schematic showing the two real-time programming logics for phytopharmacology experiments with Rubi-glutamate. Right, summary data showing the cumulative rotation degrees in mice with devices operating with the two different logics. Two-way RM ANOVA, interaction, F $(719, 5752) = 1.446$, $p < 0.0001$. $n = 5$ animals/group. Data represent mean ± SEM, Lines in the box plot show quartiles and median. $*p < 0.05$. Source data are provided as a Source Data file.

after Rubi-glutamate diffusion; Rubi-glutamate diffusion alone and control groups treated with PBS displayed no rotation bias (Fig. 5e, f). Finally, to further illustrate the technical advances introduced by real-time programmability in photopharmacology, we tested animal behavior using two distinct logics of pump/LED operation. Logic 1 involves pump operation at 4 Hz, 40% duty cycle for 120s, followed by LED operation at 20 Hz for 240s. Logic 2 involves intermingled pump and LED operation for 8 cycles. Each cycle contains 15 s pump and 30s LED operation. The switches among pump and LED operations were controlled in a real-time manner. We quantified rotational behavior in the two conditions and found that the two logics yielded different trajectories and amplitudes of behavioral outcomes (Fig. 5g). These data suggest that one can differentially and flexibly modulate neuronal activity that underlies a certain behavior response in photopharmacology experiments by controlling Pump & Light logic in real-time. Altogether, our data demonstrate the functionalities of optofluidic devices in standard independent operations in individual and grouped mice, as well as feasibility in the context of the rapidly developing field of photopharmacology.

## Discussion

The wireless, battery-free, multi-lateral optofluidic systems reported here offer numerous advanced capabilities that are not available with traditional cannulation approaches or with existing wireless technologies. Unlike metal cannulas and optical fibers made of rigid materials, the thin and soft μ-fluidic injectable probes adapt to soft bio-tissues and their micromotions, minimizing tissue damage during implantation, and reducing the activation of glial cells. The lightweight construction of the devices allows programmable multimodal drug delivery and optical stimulation without disrupting the natural behaviors of the animals during experiments. The NFC powered platform also bypasses requirements for battery power, which substantially reduces the size and weight of devices and increases

operational lifetime. The low-temperature μ-pump operation enables safe delivery of temperature-sensitive viral vectors (or other bioactive, heat-sensitive agents), and subsequent colocalized optogenetic stimulation. These features, taken together with the refillable μ-reservoirs and μ-pump systems, make this technology suitable for chronic studies such as drug efficacy tests, or in chemotherapeutics therapies that require long-term repeated drug delivery at the same target location. Moreover, with multilateral and multichannel layout configurations and full implantability, the devices offer implantation versatility to target specific regions across the central and peripheral nervous system. The flexibility of this neurotechnology creates opportunities for experimental design paradigms that involve the simultaneous stimulation of excitatory and/or inhibitory circuits of either bilateral or distal brain regions, or in multi-nervous systems from central to peripheral nervous systems. In addition, the commercially available electronics, the scalable fabrication process, and the cost effective materials greatly facilitate the potential for broad distribution of the device within the neuroscience community.

Another key advantage of the platform is in the independent, real-time multichannel control over multiple devices in a single experimental field via an intuitive and user-friendly software interface. Even the flow rate, controlled by tuning the pulse duration, can be selected in a programmable manner based on the pharmacological diffusion requirements. Moreover, independent control over multiple channels and devices enables studies of complex behavioral paradigms involving multiple animals, each with individualized stimulation parameters. This selectivity of lateralized activation of opsin-expressing neuronal groups requires effective and stable switching between channels in real time. With tunability over stimulation modalities, these platforms can support combinations of optogenetics, pharmacology, and photopharmacology. With emerging synthetic photoswitchable neuromodulators[57–59], these devices offer the ability to regulate the activation of specific types of receptors in targeted

regions of the brains of freely moving mice with high spatial and temporal resolution. Our results show that different triggering parameters (frequency and duty cycles of LEDs and pumps, time interval between different channels) can yield different behavioral results in photopharmacology experiments. The pump operation logic (e.g., continuous vs intermittent delivery) will result in distinct local drug concentration patterns that may differentially regulate the neuronal activity of the targeted brain region. Thus, the use of standard, predetermined operational parameters may limit experimental outcomes and complicate data interpretation. Real-time programmability for adjustment of parameters immediately based on behavioral results will greatly facilitate advanced exploratory neural studies and closed-loop experimental designs. This flexibility in applications is important for the rapidly developing field of photopharmacology.

Future possibilities include the integration of sensing capabilities, such as calcium sensor photometry, voltage indicators, neuromodulator sensors, and others. A closed-loop system that integrates these types of multimodal stimulation protocols with biosensing capabilities may enable supervised control of neuronal activity or organ functions, as unique capabilities for fundamental research with potential future clinical applications.

## Methods

All experimental procedures were conducted in accordance with the National Institutes of Health ethical regulations and were approved by the Institutional Animal Care and Use Committees (IACUC) of Northwestern University.

### Assembly of the circuit with both electronic base and serpentine μ-ILED probe

A sheet of double-sided Cu/Pi/Cu (18 μm/75 μm/18 μm, DuPont, Pyralux) served as a flexible printed circuit board (fPCB) substrate for the electronic base unit. An ultraviolet laser system (ProtoLaser U4, LPKF, Germany) defined and ablated the pads, interconnect traces and holes. An electroless plating process coated the interdigitated electrodes with gold (200 nm) to prevent copper oxidation under the alkaline environment of the electrolyte solution. In/Ag soldering paste (Indium Corporation) bonded the electrical components on the surface of the fPCB. Heating in a soldering machine (VP260, Manncrop) ensured fast and strong bonding. Laser ablation also defined the serpentine interconnects to the μ-ILEDs. In/Ag soldering paste (Indium Corporation) connected two blue μ-ILED (TR2227, Cree Inc.) or a blue μ-ILED and a red μ-ILED (TCE12-624, Cree Inc.) to the pads on uni-/bi-lateral probes, and this paste also connected the probe to the electronic base. A thin layer of epoxy (thickness, ~50 μm) coated the electronic components to prevent delamination. Encapsulating the entire circuit with a uniform parylene layer (14 μm; Specialty Coating Systems), followed by PDMS dip-coating layer (~200 μm), completed the process.

### NFC-enabled electronic system design

The NFC-enabled optofluidic devices relied on commercial off-the-shelf electronic components on fPCBs. Wireless power followed from the use of a double-sided planar coil (12 total loops) incorporated in the same fPCB and electrically tuned to resonate at 13.56 MHz. Two Schottky diode rectifiers in half-wave rectification configuration, a smoothing capacitor and a low dropout 3.3 voltage regulator (NCP114AMX330TBG, ON Semiconductor) provided constant voltage to produce a stable operation of the electronic circuitry. The system comprised an 8-bit low power μC (ATtiny84, Atmel Corporation) and an ISO 15693 compliant and NFC accessible EEPROM memory (M24LR04E, ST Microelectronics) for the logical control of four independent output channels and wireless communication for device control, respectively. The output channels are connected to two μ-ILEDs for optogenetics stimulation and two interdigitated electrodes (Au/Cu: thickness, 200 nm/18 μm; width: ~50 μm; gap: ~50 μm).

### Fabrication of the μ-fluidic system with passive flapper valve and probes

The passive flapper valve consisted of three layers, a bottom layer with an inlet and a ring shape channel (diameter of the outer circle: 600 μm, and diameter of the inner circle: 500 μm), a middle flexible thin film with small circular holes (diameter: 250 μm), as well as a top layer with outlet and a circular shape channel that matched with the outer ring of the bottom layer (diameter of the circle: 600 μm). Photolithography and deep reactive ion etching (STS Pegasus ICP-DRIE; SPTS Technologies, Newport, UK) patterned the design geometries on the surface of a silicon wafer. During the DRIE process, a thin layer of polymer ($CF_n$) formed on the surface for an easy mold release. Spin-casting a layer of PDMS on both molds (thickness: 100 μm, elastomer/curing agent ratio, 5:1; Sylgard 184, Dow Corning) and a polycarbonate (PC) substrate (thickness: 25 μm, elastomer/curing agent ratio, 5:1; Sylgard 184, Dow Corning), followed by baking in an oven at 75 °C for 45 min, yielded the top, bottom and middle layers. An ultraviolet laser system (ProtoLaser R, LPKF, Germany) defined the holes of the middle layer. Applying a thin film of copolymer (Microresist, PMMA 950 A3) to both top and bottom circular channels, followed by curing on a hot plate at 110 °C for 5 min, prevented the bonding of PDMS layers to each other at the coated area. Finally, the three layers were aligned and bonded sequentially under a microscope for alignment. A corona treatment and a baking process on a hot plate at 110 °C for 20 min ensured strong adhesion.

Fabrication of the μ-fluidic probes followed a similar molding method. First, photolithography and DRIE formed the channel patterns on a silicon wafer. Casting a layer of PDMS (channel layer thickness: 100 μm, cover layer thickness: 50 μm, elastomer/curing agent ratio, 10:1; Sylgard 184, Dow Corning) on the mold and a polycarbonate (PC) substrate separately, followed by curing on a hotplate at 110 °C for 30 min yielded the channel and cover layers. The channel layer was mechanically removed by a water-soluble tape and bonded with the cover layer after corona treatment for 20 s to complete the μ-fluidic channels. An ultraviolet laser system (ProtoLaser U4, LPKF, Germany) defined the channels width as ~250 μm. Bonding the μ-fluidic probes with the passive flapper valve using a similar corona treatment process completed the fabrication of the μ-fluidic system.

### Preparation of the flexible membranes for the μ-pumps

Dissolving 40 g of SIS powders (Sigma-Aldrich, MO, USA) into 40 ml toluene by 30 min sonication and 1 h stirring yielded a 1 g/ml SIS solution. Spin coating the solution onto a silicon wafer and curing it on a hotplate at 80 °C for 30 min formed a pure SIS membrane. A triple layer $Ti/Au/Ti/SiO_2$ (5 nm/50 nm/5 nm/20 nm) deposited by electron beam evaporation (AJA international) prevented gas and water vapor permeation.

### Fabrication of the μ-pump chamber and μ-reservoir

A milling machine (MDX540 CNC Mill, Roland) defined a dome shape and a cylinder shape on a COC sheet (thickness: 1 mm) for μ-reservoirs and μ-pump chambers, respectively. The side filling ports were then drilled with the same machine (diameter: 350 μm). Coating the surfaces of the μ-pump chamber and μ-reservoirs with $Ti/SiO_2$ by electron beam evaporation (5 nm/50 nm, AJA international) completed the process.

### Assembly of the device

Cylindrical μ-pump chambers patterned in the cyclic olefin copolymer were aligned to the interdigitated electrodes and bonded with a commercially available sealant (marine adhesive sealant fast cure 5200, 3M) to prevent evaporation or leakage of electrolyte. A pressure-sensitive adhesive (EL-8932EE, Adhesives Research Inc.) connected the

flexible membrane of SIS (~65 μm) on the bottom of the μ-reservoir s. The same sealant (marine adhesive sealant fast cure 5200, 3M) bonded the μ-reservoir s to the μ-pump chamber with alignment under a microscope. The inlets of the μ-fluidic system were aligned and bonded to the outlet ports of the μ-reservoirs using the same double-sided pressure-sensitive adhesive (EL-8932EE, Adhesives Research Inc.). Finally, a silicone adhesive (LO406, Loctite) adhered the μ-ILED probes and μ-fluidic channels together aligned under a microscope.

## Mechanical modeling for serpentine wires and passive flapper valves

The commercial software ABAQUS was used to optimize the geometry of the serpentine interconnects, and the shapes and material layouts of the passive flapper valve. For the electronic device, copper and PI layers were modeled by composite shell elements (S4R). The elastic modulus and Poisson's ratio values used in the simulations were 119 GPa and 0.34 for Cu, 2.5 GPa and 0.34 for PI, respectively. For the passive flapper valve, the PDMS membrane was modeled by solid hexahedron elements (C3D8R). Two types of PDMS were used in the simulations, and the elastic modulus and Poisson's ratio values were 1.2 MPa and 0.49, 2.6 MPa and 0.49, respectively. Convergence tests of the mesh size were performed to ensure accuracy.

## Micro-Particle Tracking Velocimetry (μ-PTV)

A high-resolution μ-PTV at various duty cycles using a Nikon W1 Dual Cam Spinning Disk Confocal microscope examined flow rates of the drug delivery system. The μ-PTV tracked flow particles and quantified flow characteristics in a Lagrangian frame of reference. Spherical fluorescent pink particles with diameters of 0.86 μm were used as the tracers. The microscope setup included an inverted microscope, a 95B prime Photometrics camera, LED lights, a filter cube with a 20X magnification objective lens, focused on a field of view of 30 μm × 500 μm., Operating the Photometrics camera at 640 fps with a distance-to-pixel ratio of 0.64 μm/pixel allowed for fully resolving maximum flow rates at various duty cycles. Due to the high distance-pixel ratio and relatively low number of particles N=O (10$^1$), the frame rate (640 fps) was sufficient to resolve instantaneous flow characteristics throughout the process. After preprocessing sequences to remove the background noise using ImageJ[60,61], tracers were detected at the sub-pixel level, tracked using the Hungarian algorithm, and linked with a three-frame gap closing for longer trajectories. Fourth-order B splines filtered the reconstructed trajectories to minimize the noise in the position detection and grid interpolated to produce a velocity field in the Eulerian frame of reference. Additional details on the technique can be found in previous publications[41,62].

## Flow rate modeling

To quantitatively predict the flow rate in the electrochemical devices with a target delivery volume of ~500 nL, an analytical model for the whole μ-pump system was developed; the complete derivation details can be found in Avila et al. 2021[40]. The drug delivery process can be derived from the rate form of the ideal gas law as

$$\dot{P}(V + V_0) + P\dot{V} = \dot{n}RT \tag{1}$$

where $P$ is the pressure, $V_0$ and $V$ are the initial gas volume and volume change inside the μ-pump chamber, respectively, $\dot{n} = \frac{3i}{4F}$ is the gas generation rate related to the effective current $i$ (0–1 mA) and $F$ is the Faraday's constant (96485 C mol$^{-1}$), $R$ is the ideal gas constant (8.3144 J mol$^{-1}$ K$^{-1}$), and $T$ the temperature (300 K). The force equilibrium gives

$$P = \frac{32\mu L \dot{V}}{a^4} + f(V) + P_0 \tag{2}$$

where the first term $\frac{32\mu L \dot{V}}{a^4}$ represents the μ-fluidic resistance for a square cross-section, $a$ and $L$ are the side (~30 μm), and length (~7 mm) of the delivery channel respectively, and $\mu$ is the viscosity of the fluid (0.89 Pa. s). The term $f(V)$ is the pressure needed to deform the flexible membrane leading to the volume change $V$ inside the μ-pump chamber and was determined using FEA (Supplementary Fig. 12) based on the stress-strain relationship of the flexible membrane in Supplementary Fig. 12. The third term $P_0$ is the initial environmental pressure (101.3 kPa).

The expression for the non-dimensional volume $V^*$ is

$$V^* = V^*_{SOL}(t^*) - \frac{M^*}{V^*_0\left(\frac{64h^2}{3\pi R_0^2} + \frac{P^*_0}{V^*_0}\right)^2}\left[1 - e^{-\left(\frac{64h^2}{3\pi R_0^2} + \frac{P^*_0}{V^*_0}\right)\frac{t^*}{M^*}}\right] \tag{3}$$

Where the first term $V^*_{SOL}(t^*)$ is the solution of the equation $t^* = (V^* + V^*_0)G(V^*) + V^*P^*_0$ and the second term decays exponentially with time. $P^*_0$ and $V^*_0$ are the normalized initial environmental pressure and volume, respectively, defined as $V^*_0 = \frac{V_0}{R_0^3}$, and $P^*_0 = \frac{P_0 R_0}{Eh}$, where the $R_0 \sim 1.2$ mm, $E \sim 4$ MPa, and $h \sim 150$ μm, are the flexible membrane radius, elastic modulus, thickness. $M^* = \frac{24\mu L}{a^4}\frac{iRTR_0^2}{Fh^2E^2}$ represents the normalized μ-fluidic resistance and $G(V^*) = f(V)\frac{R_0}{Eh}$ is a non-dimensional function for the membrane deformation. The non-dimensional flow rate is obtained from Eq. 3 as

$$\frac{dV^*}{dt^*} = \frac{1}{\left[f(V^*R_0^3)\frac{R_0}{Eh} + f'(V^*R_0^3)(V^* + V^*_0)\frac{R_0^4}{Eh} + P^*_0\right]} \\ - \frac{1}{V^*_0\left(\frac{64h^2}{3\pi R_0^2} + \frac{P^*_0}{V^*_0}\right)}\left[e^{-\left(\frac{64h^2}{3\pi R_0^2} + \frac{P^*_0}{V^*_0}\right)\frac{t^*}{M^*}}\right] \tag{4}$$

and the maximum flow rate in Fig. 3f was calculated from the peak value in the flow rate vs time curves in Fig. 3h for devices with different operating conditions.

## Power characterization

The power harvesting measurement was performed in an 18 × 25 cm$^2$ cage with a double-loop transmission coil and an input power of 8 W. A receiver coil connected to a potentiometer was placed into either corner or center of the cage. Measuring the voltage across the potentiometer as a function of load resistance using a battery-powered multimeter (Fluke-116, Fluke Electronics) characterized the harvested power (Supplementary Fig. 15). A custom-made nonmetallic utility that can rotate from 0–90º degrees in the middle of that cage allowed for angular measurement using this same receiver coil with an 1800 Ω shunt resistor. The voltage across the resistor with increasing angles revealed the effect of angular misalignment (Fig. 3j). For the power distribution mapping, a low drop-out 3.3 voltage regulator (NCP114AMX330TBG, ON Semiconductor) placed between the rectifier and shunt resistor ensured a constant power output. Voltage was spatially mapped with the previous load at 9 different locations and 2 heights (2.54 cm and 5.08 cm) within the experimental enclosure with an angular misalignment from 0–80º (Fig. 3k and Supplementary Fig. 17). To measure the power consumption, a 100 Ω resistor was connected in series between the rectifiers and voltage regulator. An oscilloscope (HDO4104A, Teledyne LeCroy) measured the voltage across this resistor when the device operated in different modes (Fig. 3l). The power was then calculated based on the measured voltage.

## NMR spectroscopy

Dissolving and diluting 10 mg [Ru(bpy)$_2$(PMe$_3$)GluNa$_2$](PF$_6$)$_2$ (RuBi-Glutamate) into 0.527 ml D$_2$O yielded a 10 mM Rubi-glutamate solution. Adding 10 μL 10 mM Rubi-glutamate solution into fifteen 0.6 ml microcentrifuge tubes and irradiating each tube with a blue μ-ILED (20 Hz, 40% duty cycle) for different periods of time (0, 1, 5, 10, 30 min) prepared the uncaged samples. To achieve high accuracy quantitative NMR, maleic acid serves as a reference compound of known concentration to determine the concentration conversion factor. Dissolving 22 mg maleic acid in 9.475 ml D$_2$O yields a 20 mM maleic acid solution. Adding 2 μl 20 mM maleic acid solution to each of the previously prepared tubes and diluting the solution to 0.6 ml with D$_2$O finished the sample preparation.

All 1D $^1$H NMR spectrums were obtained using 30° RF pulse (PULPROG), 2 s acquisition time (AQ), 1 s relaxation delay (D1), 256 total number of scans (NS) at room temperature (298K) through NMR HFCN600. As shown in Supplementary Fig. 22, characteristic signals of −NH$_2$ protons of Rubi-glutamate before photolysis appear at 4.06, 3.91 and 3.49 ppm (a–c), and that of the methylene protons of free glutamate appear at 3.68, 2.28, and 2.05 ppm (d–f), which match well with the spectra in the literature[63]. The characteristic signals c and d were used in Fig. 5c to determine the change of concentration for Rubi-glutamate and glutamate, respectively ($n = 3$). Glutamate concentration is not 0 at 0 min, possibly a resulting from the visible light exposure during the transferring process before the NMR spectroscopy.

## Animals

All experiments used young adult wild-type C57BL/6J mice or hybrid C57BL/6J mice on Sv129 background (~8 weeks old and 20–30 g at the start of experiments; approximately equal number of males and females were used; Jackson Labs and Charles River), maintained at ~25 °C and humidity range of 30–70%. Mice were maintained on a 12-h light/dark cycle (lights on at 6:00–7:00 AM) and fed *ad libitum*. Mice were group housed (two to four per cage) prior to surgery, after which mice were individually housed in a climate-controlled vivarium. All experimental procedures were conducted in accordance with the National Institutes of Health standards and were approved by the Institutional Animal Care and Use Committees (IACUC) of Northwestern University.

## Surgical procedures

Devices were implanted using a single standard surgical procedure[64]. Animals were anesthetized using isoflurane (3% for induction, 1.5–2% for anesthesia maintenance) and their head fur was shaved. Mice were then mounted in a stereotactic frame with a heating pad, and an incision was made down the center of the scalp to expose the skull. Burr holes for implantation of the optofluidic probes were drilled in the skull using a variable speed surgical drill. Using the attachment flag, the probes (length: 2 mm) were stereotactically lowered into the designated brain region (M2: AP = +2.1 mm, ML = ±1.0 mm, DV = −1.0 mm) at a rate of ~0.1 mm/s until appropriately positioned. The probes and the receiving coil were then affixed to the skull using dental cement to prevent further movement. Low toxicity silicone adhesive (Kwik-Sil, World precision instruments, Sarasota, FL) was used to cover the entire device to prevent physical damage to the electrical circuits. Animals were monitored and allowed to recover for several hours before being transferred back to the cage area facility for appropriate post-surgical monitoring.

For viral transduction with the μ-fluidic device, AAV8.CAG.GFP (6.7 × 10$^{12}$ GC/ml, UNC vector core, Dr. Ed Boyden) or AAV1.CaM-KIIa.hChR2(H134R).mCherry (1.2 × 10$^{13}$ GC/ml, Addgene viral prep # 26975-AAV1, Dr. Karl Deisseroth) were loaded into μ-reservoirs prior to device implantation (~500 nL). After animals recovered from anesthesia, μ-fluidic channels were activated (250 ms, 50% duty cycles). For photopharmacology with Rubi-glutamate, 10 mM Rubi-glutamate solution (~500 nL) was loaded into μ-reservoirs prior to the behavioral experiments. Animals were allowed to recover for at least 3 days before any behavioral experiments were conducted.

## Histology

Coronal brain sections, 60 μm thick, were prepared using vibratome (VT1000S, Leica Biosystems, Buffalo Grove, IL). For evaluation of biocompatibility, rabbit anti-GFAP (1:1000, ab7260, Abcam, Cambridge, United Kingdom), and rabbit anti-IBA1 (1:1000, ab178846, Abcam) were used as described in previous studies[19,65]. On the following day, tissues were rinsed three times with PBS, reacted with anti-rabbit Alexa Fluor 647 secondary antibody (1:500, Thermo Fisher, Waltham, MA) for 2 hr at RT, and rinsed again three times in PBS. Sections were mounted on Superfrost Plus slides (Thermo Fisher, Waltham, MA), air dried, and cover slipped under glycerol:TBS (9:1) with Hoechst 33342 (2.5 μg/ml, Thermo Fisher Scientific). Analysis was carried out in FIJI[60,61] using autothresholding and area measurement scripts. The same analysis parameters were applied across all regions of interest. The values for a specific implantation were calculated from the average of two individual ROIs.

## Electrophysiology

Coronal brain slice preparation was modified from previously published procedures[47,66]. Animals were deeply anesthetized by inhalation of isoflurane, followed by a transcardial perfusion with ice-cold, oxygenated artificial cerebrospinal fluid (ACSF) containing (in mM) 127 NaCl, 2.5 KCl, 25 NaHCO$_3$, 1.25 NaH$_2$PO$_4$, 2.0 CaCl$_2$, 1.0 MgCl$_2$, and 25 glucose (osmolarity 310 mOsm/L). After perfusion, the brain was rapidly removed, and immersed in ice-cold ACSF equilibrated with 95% O$_2$/5%CO$_2$. Tissue was blocked and transferred to a slicing chamber containing ice-cold ACSF, supported by a small block of 4% agar (Sigma-Aldrich). Bilateral 250 μm-thick mPFC slices were cut on a Leica VT1000S (Leica Biosystems, Buffalo Grove, IL) in a rostro-caudal direction and transferred into a holding chamber with ACSF, equilibrated with 95%O$_2$/5%CO$_2$. Slices were incubated at 34 °C for 30 min prior to electrophysiological recording. Slices were transferred to a recording chamber perfused with oxygenated ACSF at a flow rate of 2–4 ml/min at room temperature.

Voltage clamp whole-cell recordings were obtained from neurons visualized under infrared DIC contrast video microscopy using patch pipettes of ~2–5 MΩ resistance. M2 pyramidal neurons were visually identified. To uncage Rubi-glutamate, light pulses (460 nm, 20 ms) were delivered every 30 s at the recording site using whole-field illumination through a 60X water-immersion objective (Olympus, Tokyo, Japan) with a PE4000 CoolLED illumination system (CoolLED Ltd., Andover, UK). Recording electrodes contained the following (in mM): 120 CsMeSO$_4$, 15 CsCl, 10 HEPES, 10 Na-phosphocreatine, 2 MgATP, 0.3 NaGTP, 10 QX314, and 1 EGTA (pH 7.2–7.3, ~295 mOsm/L). Recordings were made using 700B amplifiers (Axon Instruments, Union City, CA); data were sampled at 10 kHz and filtered at 4 kHz with a MATLAB-based acquisition script (MathWorks, Natick, MA).

## Experimental setup and operation of the devices for behavioral studies

A customized GUI implemented in MATLAB 2021 (Mathworks) provides individual control of multiple devices in the same experimental enclosure in real time. The host computer communicates with the RF power distribution box (PDC box, NeuroLux Inc) using an RS232 communication protocol (an RS232 to USB converter was used). The PDC box, which supports the ISO 15693 NFC communication protocol, is the link to gain access to the devices inside the experimental enclosure. Two-loop coil antenna, tuned to 13.56 MHz, around the experimental enclosure (18 x 25 cm$^2$) provided wireless power transfer to the devices as well as communication. Using the GUI, the user had control of different parameters of the devices such as channel

activation (ON/OFF commands), mode of operation (single or dual channel) and stimulation parameters (frequency and duty cycles). The version of the GUI used here for the behavioral studies allows addressing four devices in the experimental arena. Although not implemented here, the on-demand activation of a target device via external signal controls such as transistor-transistor logic (TTL) represents a straightforward and useful system upgrade.

Loading the electrolyte (50 mM potassium hydroxide, KOH) into the μ-pump chamber through the bottom refilling ports with a 30G needle attached to a 1 ml syringe prepares the electrolysis actuator. A UMP3 micropump (World Precision Instruments) fills the drug chamber with the virus or Rubi-glutamate via a pulled glass pipette through the top refilling ports. A stiff adhesive layer (EL-8932EE, Adhesives Research Inc.) covers the top μ-fluidic channels to prevent damage from mice and water evaporation. Sealing the ports with a removable layer of silicone elastomer (Kwik-sil; World Precision Instruments) finishes the device preparation before implantation.

### Behavioral experiments

Animals were placed in a $15 \times 15$ cm$^2$ arena wrapped with an antenna connected to a Neurolux system. After a 2-min-long acclimatization period, the motor behaviors of animals were recorded. μ-ILED were programmed to deliver 2 ms-long pulses at 20 Hz for optogenetics, and 20 ms-long pulses at 20 Hz for photolysis of Rubi-glutamate. μ-pumps were programmed at 250 ms with 30–50% duty cycles for delivery of Rubi-glutamate. Videos were recorded using a Raspberry Pi camera with a resolution of $1280 \times 720$ at 25 fps. The snout, ears, hind legs, and the base of the tail of each mouse were labeled using Deeplabcut[45]. For each video, one frame per second was automatically selected in a uniform manner to be manually labeled, generating the training and testing dataset for the Deeplabcut algorithm. The test error with a p-cutoff value of 0.6 is 4.85 pixels. To quantify the rotational behavior of mice, we selected one automatically labeled frame per 0.5 s. The 'head' position was defined as the centroid of the coordinates of the snout and ears, while the 'back' position was defined by the centroid of the coordinates of two legs and the base of the tail. The body vectors were created from 'back' to 'head' positions for all labeled frames. Two body vectors separated by 0.5 s were designated as $v_1$ $(x_1, y_1)$ and $v_2$ $(x_2, y_2)$ respectively, and the angle mice rotated in this period is calculated as follows:

$$\text{angle} = \arctan2(x_1y_2 - y_1x_2, x_1x_2 + y_1y_2)$$

where $\arctan2(y, x)$ corresponds to the arc tangent method in Python's math package (Python 3.9, Python Software Foundation) with $0-2\pi$ radians domain. The angle was converted from radian to degree (°). The degrees rotated per 0.5 s and cumulative rotated degree were calculated for each experiment.

### MicroCT imaging

Animals were euthanized and placed inside a preclinical microPET/CT imaging system (Mediso nanoScan PET/CT). Data were acquired with 'medium' magnification, 33 μm focal spot, $1 \times 1$ binning, with 720 projection views over a full circle, with a 300 ms exposure time, and 70 kVp (where kVp is peak voltage). The projection data were reconstructed with a voxel size of 34 μm using filtered (Butterworth filter) back-projection software from Mediso Nucline (v2.01). Reconstructed images were filtered with Amira v2020.2 (Thermo-Fisher) using a non-local means filter. Filtered data were segmented in Amira to highlight the device. Finally, a 3D surface rendering was made using different colormaps for the skeleton and device.

### Statistics and reproducibility

Group statistical analyses were performed using OriginPro2019 and GraphPad Prism 7 software (GraphPad, LaJolla, CA). For N sizes, the number of trials, samples, and animals is provided. No statistical methods were used to pre-determine sample sizes, but our sample sizes are similar to those reported in the previous publications[19]. All samples were randomly assigned to experimental groups. Analyses were performed blindly to experimental conditions. Data from failed devices were excluded from the analysis. Data are expressed as mean ± SEM, individual plots, or box plots showing quartiles and median. For two-group comparisons, statistical significance was determined by two-tailed Student's $t$-tests or Mann–Whitney test. For multiple group comparisons, analysis of variance (ANOVA) tests was used, followed by post hoc analyses. $P < 0.05$ was considered statistically significant. All representative data were from experiments that were independently repeated for at least five times with similar results. Schematics were created with BioRender.com.

### Reporting summary

Further information on research design is available in the Nature Research Reporting Summary linked to this article.

## Data availability

Source data are provided with this paper. Raw video data generated during the current study are available from the corresponding author on reasonable request. The numerical data generated and analyzed during the current study are available at https://github.com/A-VazquezGuardado/Smart_NFC_Optofluidics. Source data are provided with this paper.

## Code availability

All computer code and customized software generated during and/or used in the current study is available at https://github.com/A-VazquezGuardado/Smart_NFC_Optofluidics (https://doi.org/10.5281/zenodo.6946429).

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

## Acknowledgements

This work made use of the NUFAB facility of Northwestern University's NUANCE Center, which has received support from the SHyNE Resource (NSF ECCS-2025633), the IIN, and Northwestern's MRSEC program (NSF DMR-1720139). Imaging work was performed at the Northwestern University Center for Advanced Microscopy generously supported by NCI CCSG P30 CA060553 awarded to the Robert H Lurie Comprehensive Cancer Center. This work also made use of the MatCI Facility which receives support from the MRSEC Program (NSF DMR- 1720139) of the Materials Research Center at Northwestern University. Funding for the electronics and μ-fluidic systems was provided by the Querrey-Simpson Institute for Bioelectronics. Y.K. is supported by the NIH R01MH117111 and R01NS107539, One Mind Nick LeDeit Rising Star Award, Rita Allen Foundation Scholar Award, the Searle Scholar Award, and Beckman Young Investigator Award. M.W is supported as an affiliate fellow of the NIH T32 AG20506 and 2021 Christina Enroth-Cugell and David Cugell Fellow. Y.Z. is supported by the University of Connecticut start-up fund, NIH RF1NS118287, NIH R42MH116525, and NIH R61DA051489. R.A. acknowledges support from the National Science Foundation Graduate Research Fellowship (NSF grant number 1842165) and Ford Foundation Predoctoral Fellowship. NeuroLux acknowledges support from the NIH 5R42MH116525-03. S.M. is supported by the German Research Foundation (DFG) Postdoctoral Fellowship and The Ellen R. and Melvin J. Gordon Center Fellowship.

## Author contributions

Y.W., M.W., S.M., B.L.S., Y.K. and J.A.R. conceived the project; Y.W., M.W., A.V-G., J.K., Y.Z., Y.K. and J.A.R. designed the system and experimental methods; A.V-G. developed the software; Y.W., M.W., A.V-G., J.K., X.Z., J.T.K., Y.Y., S.M., Y.B., L.M., H.G., L.H., Z.H., C.R.H., E.K., F.L. investigated and performed the experiments; Y.W., M.W., A.V-G., J.K., X.Z., R.A., J.T.K., Y.B., H.Y., L.M., L.H., E.A.W. conducted the data analysis; Y.W., M.W., A.V-G., J.K., R.A., J.T.L., Y.D., Y.B., L.M. wrote the original draft of the manuscript; Y.W., M.W., A.V-G., Y.K. and J.A.R. reviewed and edited the manuscript; Y.W., M.W., A.V-G. and J.K. ; B.L.S., L.P.C., Y.H., Y.K. and J.A.R. supervised the project; Y.W. managed the project; Y.K., A.R.B. and J.A.R. acquired the funding.

## Competing interests

J.A.R. and A.R.B. are founders in a company, Neurolux, Inc., that offers related technology products to the neuroscience community. The remaining authors declare no competing interests.

## Additional information

[1]Department of Materials Science and Engineering, Northwestern University, Evanston, IL, USA. [2]Querrey Simpson Institute for Bioelectronics, Northwestern University, Evanston, IL, USA. [3]Center for Bio-Integrated Electronics, Northwestern University, Evanston, IL, USA. [4]Department of Neurobiology, Northwestern University, Evanston, IL, USA. [5]Center for Bionics of Biomedical Research Institute, Korea Institute of Science and Technology, Seoul 02792, Republic of Korea. [6]Department of Mechanical Engineering, Northwestern University, Evanston, IL, USA. [7]State Key Laboratory of Mechanical System and Vibration, Shanghai Jiao Tong University, Shanghai, China. [8]Neurolux Inc, Northfield, IL, USA. [9]Department of Neurobiology, Howard Hughes Medical Institute, Harvard Medical School, 220 Longwood Ave, Boston, MA 02115, USA. [10]Polymer Program, Institute of Materials Science, University of Connecticut, Storrs, CT 06269, US. [11]Department of Biomedical Engineering, University of Connecticut, Storrs, CT 06269, US. [12]Mechanical Science and Engineering Department, University of Illinois, Urbana, IL, USA. [13]Department of Molecular Biosciences, Northwestern University, Evanston, IL, USA. [14]Center for Advanced Molecular Imaging, Radiology, and Biomedical Engineering, Northwestern University, Evanston, IL 60208, USA. [15]Department of Civil and Environmental Engineering, Northwestern University, Evanston, IL, USA. [16]Chemistry of Life Processes Institutes, Northwestern University, Evanston, IL, USA. [17]Department of Biomedical Engineering, Northwestern University, Evanston, IL, USA. [18]Simpson Querrey Institute & Feinberg Medical School, Northwestern University, Evanston, IL, USA.

[19]Department of Chemistry, Northwestern University, Evanston, IL, USA. [20]Department of Neurological Surgery, Northwestern University, Evanston, IL, USA. [21]Department of Electrical and Computer Engineering, Northwestern University, Evanston, IL, USA. [22]Department of Computer Science, Northwestern University, Evanston, IL, USA. [23]These authors contributed equally: Yixin Wu, Mingzheng Wu, Abraham Vázquez-Guardado, Joohee Kim.
✉e-mail: y-huang@northwestern.edu; yevgenia.kozorovitskiy@northwestern.edu; jrogers@northwestern.edu

