## [Peer Review File · Nature Communications]

Wireless multi-lateral optofluidic microsystems for real-time programmable optogenetics and photopharmacologyREVIEWER COMMENTS

Reviewer #1 (Remarks to the Author):

The paper of Wu et al. describes a novel wireless, implantable system for multi-wavelength optogenetics, fluid and gene delivery, and photopharmacology. Additional advantages and desirable features include: interconnect flexibility, real-time programmability of multiple devices through a graphical interface, moderate cost, versatility of the unilateral Vs. bilateral configurations, flow control valve, and refillable reservoirs.

To functionality the device and light and fluid delivery performance during different mode of operations are rigorously and extensively characterized through theoretical modeling, benchtop, and in vivo testing.

The device is innovative and its impact for the broad neuroscience and neuroengineering communities is clear. The experimental methods are thoroughly described and claims are well supported by the data.

Few points need to be addressed before the manuscript can be accepted for publication:

1. Introduction: add citations of previous works from the Anikeeva's group on one-step optogenetics and photopharmacology through multimodal fiber implants (see for example: Park et al. Nat neuroscience 20, 2017, Frank et al. ACS Chemical Neuroscience, 11, 2020)
2. Figure 2: Specify the flow and u-LED operation conditions used in the panels. Also, panels d and h appear to show backflow (i.e., the flow of infusate along the implant tract). Backflow can be detrimental for the device operation, as it can lead to unwanted off-target delivery and side-effects. The authors should evaluate - at least in phantoms - relevant delivery parameters, such as volume distribution, distribution ratio, flow rate limits and rate of backflow occurrence at varying flow rates.
3. The definition of total tissue damage is not clear. Does it correspond to the device footprint in the tissue? Also, can the authors elaborate on the finding of reduced microglia activation, but same levels of astrocyte response compared to the 500 um cannula?
4. What is the final thickness of an assembled device with the microfluidic+1-2uLEDs?

Reviewer #2 (Remarks to the Author):

General comment:

The authors presented a systematic work of device fabrication, characterization and implementation of the wirelessly operated optofluidic system with colocalized drug delivery and optical stimulation. Apart from the off-the-shelf electronic components, the author developed and optimized key items of the coiled receiving antenna, electrolysis μ -pump and μ -reservoir, microfluidic check valves and μ -LEDs. Each component has been characterized with great depth to satisfy the in vivo implantation on freely-moving mice. The rationally implemented animal experiments provide substantial evidence that future optogenetic or photopharmacologic manipulation of animal behaviors can be easily addressable through a remote-control manner. Acceptance of this manuscript is strongly encouraged after addressing the following very minor questions.

Additional questions:

1. The authors claimed that gold coating could prevent electrode corrosion from the electrolyte and electrolysis reactions. However, explanations should be provided why the electrode still degraded significantly according to Fig. 8c. In addition, the authors could compare the images to those without gold coating to support this claim.
2. The authors implemented a multilayer coating of Ti/Au/Ti/SiO₂ on the separator membrane to reduce gas permeation. Are there previous reports or experimental comparisons to support the

effectiveness of the coating? Is this complicated configuration necessary and how is it compared with the single-layer coating?

3. In the biocompatibility section, how long after implantation was the immunostaining conducted?

Reviewer #3 (Remarks to the Author):

The manuscript describes a wireless optofluidic system that allows to infuse liquid and deliver light to target brain regions. The authors showed the capability of the multilateral optofluidic system through in-vivo experiments with freely moving mice. Also, they showed the system could be utilized for photopharmacology. The in-vivo experiments were well performed and showed the functions of the system.

However, I have some concerns about the technical advances of the system compared to previous wireless optofluidic systems. There have been several wireless optofluidic systems, as the authors describe in Table 1. The previous systems also enabled wireless delivery of liquid and light with multiple remote-control capabilities. The difference of this system is that it has two probes and injects fluid, and delivers light to two brain regions simultaneously. Also, the check valve integrated into the system potentially prevents leakage and backflow. I agree that integrating two probes in a single platform is not trivial. However, I could not find any technical hurdles in the manuscript.

Also, the authors implemented the real-time programmability of the proposed system and characterized the flow rate. However, real-time programmability was not used during the in-vivo experiment. The in-vivo experiments in Fig. 4 and Fig. 5 could have been done with previous wireless optofluidic systems, which do not have flow rate modulation. At least, the experiment which can show the advantage of real-time programmability of pumps needs to be conducted.

During the implantation of the device, two probes were implanted separately. I think this procedure induces positioning error for targeting the brain region. It is required to estimate the positioning accuracy of implantation.

The authors described in line 248 that “the flow rate must remain below levels that could induce damage to adjacent healthy tissue.” What is the flow rate to avoid the damage?

It seems that the flow rate was measured with the probe implanted in agarose gel. However, the flow rate might be different if it were measured with the probe implanted in a brain. How much difference would be expected?

In Supplementary Figure 13, the flow rate does not seem to be controlled according to duty cycles. Especially, the flow rate at 30% duty cycle is higher than that of 40% duty cycle, and the abrupt increases of the flow rate found in the 20% duty cycle and 40% duty cycle might induce tissue damage. How many measurements have you done for this experiment in each condition? It is required to mention the results with any possible reasons. If it is due to a less filled solution in the reservoir, the experiment needs to be conducted with a fully loaded reservoir or elimination of bubbles in the reservoir.

It is required to specify the dimension of the probe implanted in the brain, such as total thickness, width, and length.

The authors showed only an area of astrocyte and microglia in the histology data. However, the intensity information is also important to compare the biocompatibility. It is required to add intensities of fluorescence signal according to the distance from edges.

Summary of revisions

We would like to thank the Reviewers for support and excellent suggestions. We have modified the manuscript and provided new data to address all the Reviewers' concerns.

This revision includes the following new figure panels: Fig. 5g. Supplementary Fig. 4, Supplementary Fig. 14g, Supplementary Fig. 19c, d, e. Supplementary Fig. 20.

Summary of changes:

1. We included a new in vivo photopharmacology experiment and additional discussion to demonstrate the advantage of real-time programmability.
2. We performed new experiments and analyses to estimate the positioning accuracy of individually implantable probes, and we introduce a new probe design to increase accuracy.
3. We performed additional characterizations for backflow in hydrogel when operating the devices under different duty cycles and found no backflow when duty cycles are equal to or lower than 50%.
4. We compared the interdigitated electrodes with and without gold coating, demonstrating that gold coating can avoid corrosion effectively during electrochemical reaction.

Major changes to the manuscript text are highlighted in red color. Line numbers are also referenced throughout responses to specific comments.

Response to Reviewer 1

Comment: The paper of Wu et al. describes a novel wireless, implantable system for multi-wavelength optogenetics, fluid and gene delivery, and photopharmacology. Additional advantages and desirable features include: interconnect flexibility, real-time programmability of multiple devices through a graphical interface, moderate cost, versatility of the unilateral Vs. bilateral configurations, flow control valve, and refillable reservoirs.

To functionality the device and light and fluid delivery performance during different mode of operations are rigorously and extensively characterized through theoretical modeling, benchtop, and in vivo testing.

The device is innovative and its impact for the broad neuroscience and neuroengineering communities is clear. The experimental methods are thoroughly described and claims are well supported by the data. Few points need to be addressed before the manuscript can be accepted for publication:

1. Introduction: add citations of previous works from the Anikeeva's group on one-step optogenetics and photopharmacology through multimodal fiber implants (see for example: Park et al. Nat neuroscience 20, 2017, Frank et al. ACS Chemical Neuroscience, 11, 2020)

Response: We thank the reviewer for their positive comments and their input. We added the suggested references to the manuscript.

Modifications to the manuscript:

Page 4, line 82:

Ref 21: Park, S., et al. "One-step optogenetics with multifunctional flexible polymer fibers." Nature neuroscience 20(4), 612-619. (2017)

Ref 22: Frank, J. A., et al. "In vivo photopharmacology enabled by multifunctional fibers." ACS chemical neuroscience 11(22), 3802-3813. (2020)

Comment: 2. Figure 2: Specify the flow and μ -LED operation conditions used in the panels. Also, panels d and h appear to show backflow (i.e., the flow of infusate along the implant tract). Backflow can be detrimental for the device operation, as it can lead to unwanted off-target delivery and side-effects. The authors should evaluate - at least in phantoms - relevant delivery parameters, such as volume distribution, distribution ratio, flow rate limits and rate of backflow occurrence at varying flow rates

Response: We agree with the reviewer that evaluation of backflow during fluidic delivery is essential. We performed additional experiments and added supplementary figures to evaluate the backflow issue at various duty cycles in brain phantoms. Our data shows that low duty cycle operation can reduce the backflow effect.

We specified the pump and the LED parameters in the Fig 2 legend.

Supplementary Figure 4 Photos of the delivery of dyed aqueous solution in hydrogel with optofluidic devices operated at (a) 10% duty cycle, (b) 50% duty cycle, (c) 100% duty cycle, 4Hz frequency. Scale bar: 0.5 mm. The photo after 0, 1 and 2 min of delivery demonstrates that low duty cycle delivery allows for a lower flow rate, which can reduce the backflow effect.

Modifications to the manuscript:

We added the following sentences and supplementary figures to the manuscript.

Fig 2 line 3: "Photographs that demonstrate bilateral optofluidic devices operated at single mode and dual mode for both optical stimulation (4Hz, 50% duty cycle) and fluidic delivery (4Hz, 50% duty cycle)."

Fig 2 line 5: "(f) Photograph that demonstrates unilateral bi-channel optofluidic devices with single/dual two-wavelength optical stimulation (4Hz, 100% duty cycle)."

Fig 2 line 7: “(h) Photograph that demonstrates a unilateral bi-channel optofluidic with single/dual modes of liquid infusion in hydrogel phantom brain tissue (4Hz, 100% duty cycle).”

Fig 2 line 9: “(j) Photographs of fluorophore infusion through μ -pump actuation (left, 4Hz, 50% duty cycle) and green fluorescence emission of the fluorophore after stimulated by blue μ -ILEDs (right, 4Hz, 50% duty cycle).”

Page 12, line 252: “..., but it must remain below levels that could induce damage to adjacent healthy tissue and avoid backflow along the insertion tract to eliminate the off-target effect³⁹ (**Supplementary Fig. 13**).”

Comment: 3. The definition of total tissue damage is not clear. Does it correspond to the device footprint in the tissue? Also, can the authors elaborate on the finding of reduced microglia activation, but same levels of astrocyte response compared to the 500 μm cannula?

Response: We measured the area of damage in cross sections of brain tissue. We added illustration showing the measurement of damage area and changed the term ‘Total tissue damage’ to ‘Damage area in horizontal slices’ to reduce ambiguity. The area of damage corresponds to the geometry of the implantable probes and cannula. We added potential explanations of reduced microglia activation in the discussion section. For example, the larger amount of brain tissue damage induced by the 500 μm cannula may result in stronger microglial activation¹.

Modifications to the manuscript:

Page 15, line 320: “(Notably, significantly lower activation of microglial cells (IBA1) and smaller damage area in horizontal brain slices was observed in subjects implanted with μ -fluidic probes, compared to the cannula (Supplementary Fig. 18). Lower microglial activation may potentially result from decreased amount of brain tissue damage with the probes.)”

Comment: 4. What is the final thickness of an assembled device with the microfluidic+1-2 μLEDs ?

Response: The total thickness of the probe is $\sim 350 \mu\text{m}$ includes Parylene C $28 \mu\text{m}$ ($14 \mu\text{m}$ each side), Cu-Pi electrodes $93 \mu\text{m}$, μ -LED $50 \mu\text{m}$, μ -fluidic $150 \mu\text{m}$ and epoxy $\sim 50 \mu\text{m}$. We have added the parameters of the probe in the manuscript.

Modifications to the manuscript:

Page 6, line 119: “(Probe total thickness: $\sim 350 \mu\text{m}$, Distance...)”

Response to Reviewer 2

Comment: General comment:

The authors presented a systematic work of device fabrication, characterization and implementation of the wirelessly operated optofluidic system with colocalized drug delivery and optical stimulation. Apart from the off-the-shelf electronic components, the author developed and optimized key items of the coiled receiving antenna, electrolysis μ -pump and μ -reservoir, microfluidic check valves and μ -LEDs. Each component has been characterized with great depth to satisfy the in vivo implantation on freely-moving mice. The rationally implemented animal experiments provide substantial evidence that future optogenetic or photopharmacologic manipulation of animal behaviors can be easily addressable through a remote-control manner. Acceptance of this manuscript is strongly encouraged after addressing the following very minor questions.

Response: We thank the reviewer for appreciating this work and for recommending acceptance.

Comment: 1. The authors claimed that gold coating could prevent electrode corrosion from the electrolyte and electrolysis reactions. However, explanations should be provided why the electrode still degraded significantly according to Fig. 8c. In addition, the authors could compare the images to those without gold coating to support this claim.

Response: We agree with the Reviewer that additional experiments are needed to further support our claim that electrodes can be significantly damaged without the gold coating. We performed experiments comparing the degradation of electrodes with and without gold coating during operation. Our data show that without the gold coating, the electrodes degrade significantly after one cycle of operation (4 Hz, 50% duty cycle, 2 min). Gold coated electrodes show little corrosion compared to ones without coating.

Supplementary Figure 8 (d) Optical images of copper electrochemical electrodes without gold coating after 0, 1, 4 cycles of use in electrochemical pumping. Scale bar: 0.5 mm.

Modifications to the manuscript: We added the following sentences and supplementary figures for pure copper electrodes to the manuscript.

Page 10, line 217: “The gold coating on the electrodes prevents corrosion from the electrolyte and the electrolysis process (Supplementary Fig. 9)”.

Comment: 2. The authors implemented a multilayer coating of Ti/Au/Ti/SiO₂ on the separator membrane to reduce gas permeation. Are there previous reports or experimental comparisons to support the effectiveness of the coating? Is this complicated configuration necessary and how is it compared with the single-layer coating?

Response: We thank the reviewer for this comment. In Figure 3b, we compared the relaxation of the membrane

with/without the coating, showing the effectiveness of the coating in reducing gas permeation. The multi-layer coating is necessary for strong adhesion of the gold coating to the polymer substrate and silicon dioxide layer. The SiO₂ layer renders a hydrophilic surface to avoid bubble formation in the drug reservoirs

Modifications to the manuscript: We added the following to the manuscript.

Page 10, line 225: “Without the gold coating, the film can quickly relax back to its original geometry due to transmembrane permeation of trapped gas and to recombination of oxygen and hydrogen in the electrolysis reaction. The titanium layers promote adhesion between the gold and polymer substrate as well as the gold and SiO₂ layer. The SiO₂ renders a hydrophilic surface to avoid bubble formation in the drug reservoirs.”

Comment: 3. In the biocompatibility section, how long after implantation was the immunostaining conducted?

Response: We extracted the brains and performed immunostaining 7 days after implantation. We added the information to the corresponding result section.

Modifications to the manuscript:

Page 15, line 318: “Mouse brains were extracted seven days after implantation and examined with immunostaining.”

Response to Reviewer 3

Comment: The manuscript describes a wireless optofluidic system that allows to infuse liquid and deliver light to target brain regions. The authors showed the capability of the multilateral optofluidic system through in-vivo experiments with freely moving mice. Also, they showed the system could be utilized for photopharmacology. The in-vivo experiments were well performed and showed the functions of the system.

However, I have some concerns about the technical advances of the system compared to previous wireless optofluidic systems. There have been several wireless optofluidic systems, as the authors describe in Table 1. The previous systems also enabled wireless delivery of liquid and light with multiple remote-control capabilities. The difference of this system is that it has two probes and injects fluid, and delivers light to two brain regions simultaneously. Also, the check valve integrated into the system potentially prevents leakage and backflow. I agree that integrating two probes in a single platform is not trivial. However, I could not find any technical hurdles in the manuscript.

Response: We thank the Reviewer for the comments. We edited the manuscript and performed new experiments to highlight the technical advances of the current system.

Although previous wireless optofluidic systems demonstrated functionalities in fluidic delivery with remote control, they generally lack the capabilities to fully control all relevant parameters of fluidic delivery and optical illumination in precise and programmable manner. These control capabilities are essential for researchers to utilize the wireless optofluidic system in complex animal studies that involve multiple brain regions in individual animals or social groups.

The technical advance of our system includes multilateral and multichannel layout with advanced electronic design, allowing programmable control over probes in either single or grouped animals with easy implantation process, as well as precise control of flow dynamics via valves and real-time programmability. Altogether, this work overcomes the major limitations of previously described devices.

Comment 1: Also, the authors implemented the real-time programmability of the proposed system and characterized the flow rate. However, real-time programmability was not used during the in-vivo experiment. The in-vivo experiments in Fig. 4 and Fig. 5 could have been done with previous wireless optofluidic systems, which do not have flow rate modulation. At least, the experiment which can show the advantage of real-time programmability of pumps needs to be conducted.

Response: We thank the reviewer for this comment. We agree that the necessity and advantage of real-time programmability of the pumps required further demonstration. We performed additional behavioral experiments and made clarifications in the manuscript to address this concern.

To illustrate the technical advances introduced by real-time programmability, we designed two distinct logics of pump/LED operation in a photopharmacology experiment. Logic 1 involves pump operation at 4Hz, 40% duty cycle for 120s, followed by LED operation at 20Hz for 240s. Logic 2 involves intermingled pump&LED operation for 8 cycles. Each cycle contains 15s pump and 30s LED operation. The switches among pump and LED operations were controlled in a real-time manner. We quantified rotational behavior in the two conditions and found that the two logics yielded different trajectories and amplitudes of behavioral outcomes. These data demonstrate that one can differentially and flexibly modulate neuronal activity that underlies a certain behavior response in photopharmacology experiments by controlling Pump & Light logic in real-time. This operation is enabled by precise real-time control of individual elements that has not been previously achieved.

In addition, we would like to clarify that the in vivo experiments in Fig. 4 and Fig. 5 cannot be performed efficiently with previously reported wireless optofluidic systems. Regarding Fig. 4, first, the bilateral motor behavior

experiments require multi-lateral probes of a type that have not been described in previous designs. The selectivity of lateralized activation of opsin-expressing neuronal groups requires effective and stable switching between channels in real-time. Second, multilateral operation in groups requires not only multi-lateral probes but also real-time selectivity and programmability of a device within a group of devices. This capability is also impossible with any previous optofluidic platforms.

Regarding new data in Fig. 5, we showed in the new experiments that different triggering parameters (frequency and duty cycles of LEDs and pumps, time interval between different channels) can yield different behavior results in photopharmacology experiments. Thus, pre-determined operational parameters (as reported in many previous designs) may severely limit experimental outcomes and complicate data interpretation. Real-time programmability for adjustment of parameters immediately based on behavioral results will greatly facilitate advanced exploratory neural studies, not possible with previous devices. This new flexibility in applications is important for the rapidly developing field of photopharmacology.

Modifications to the manuscript: We added additional text in Results and Discussion.

Page 18, line 399: “(Finally, to further illustrate the technical advances ...)”

Page 20, line 440: “(This selectivity of lateralized activation of opsin-expressing neuronal groups requires...)”

Page 20, line 446: “(Our results showed that different triggering parameters ...)”

Comment 2: During the implantation of the device, two probes were implanted separately. I think this procedure induces positioning error for targeting the brain region. It is required to estimate the positioning accuracy of implantation.

Response: We performed new experiments to estimate positional accuracy for two separately implanted probes and a single rigidly joined probe. We targeted the same brain region in the two brain hemispheres using (1) two separately implantable probes (Flexible design), and (2) an integrated probe with two tips (Fixed design). After implantation, we fixed the brain tissue and performed histological assessments of the displacement between two probe tips. We set one probe location as the reference (X_0, Y_0), and measured the distance from center point in the other hemisphere (XY in absolute value) of the other probe location. Our result showed that the displacement (Mean \pm SEM) in the X-axis (medial-lateral axis) is $42.05 \pm 10.42 \mu\text{m}$ in the fixed design and $153.6 \pm 48.24 \mu\text{m}$ in the flexible design, while the displacement in the Y-axis (anterior-posterior axis) is $108.1 \pm 18.26 \mu\text{m}$ in the fixed design and $84.14 \pm 26.14 \mu\text{m}$ in the flexible design. Therefore, we introduced the fixed probe option to reduce the surgical complexity based on user preference, although the individually implantable probes provide greater flexibility to target different combinations of brain regions of interest.

Modifications to the manuscript:

Page 15, line 326: “(To estimate positional accuracy for the flexible optofluidic probes...)”

Comment 3: The authors described in line 248 that “the flow rate must remain below levels that could induce damage to adjacent healthy tissue.” What is the flow rate to avoid the damage?

Response: We thank the reviewer for the comment. High flow rate will lead to backflow, and associated tissue damage and off-target delivery. Previous work shows that smaller catheters and lower flow rates can avoid backflow. For instance, in rat gray matter, a 32-gauge catheter with a flow rate below $0.5 \mu\text{L}/\text{min}$ can diminish backflow (Morrison, P. F., et al. American Journal of Physiology-Regulatory, Integrative and Comparative Physiology 277.4, 1999). A flow rate below $0.5 \mu\text{L}/\text{min}$ is suggested to avoid damage.

Modifications to the manuscript: We added additional text and reference to the manuscript.

Page 12, line 254: “but it must remain below levels that could induce backflow, which leads to off-target delivery and side effects.³⁹”

Ref 39: Morrison, P. F., et al. "Focal delivery during direct infusion to brain: role of flow rate, catheter diameter, and tissue mechanics." *American Journal of Physiology-Regulatory, Integrative and Comparative Physiology* 277(4), R1218-R1229. 1999.

Comment 4: It seems that the flow rate was measured with the probe implanted in agarose gel. However, the flow rate might be different if it were measured with the probe implanted in a brain. How much difference would be expected?

Response: We agree that fluidic delivery in agarose gel can mimic that in a brain, with some differences. Previous literature shows that pressure profiles and volumes of distribution of low-molecular-weight dyes and imaging contrast agents in a 0.6% agarose gel and an in vivo porcine model are strikingly similar, demonstrating that the gel is a reasonable facsimile of the brain for flow rate estimates (Chen, Z. J., et al. *Journal of neurosurgery* 101(2), 314-322. 2004.). The difference lies in the uptake and active clearance of the infused agents in the brain, as well as the inhomogeneous and highly anisotropic nature of brain tissue compared to gel. Because of these factors, differences can appear based on the target region and specific animal species and age. In spite of these differences, the agarose gel model is useful as a simplified version of the brain for preliminary estimates of flow.

Modifications to the manuscript: We discussed the expected differences in the manuscript.

Page 13, line 278: “(Notably, the fluidic characteristics may differ in the brain because of ...)”

Comment 5: In Supplementary Figure 13, the flow rate does not seem to be controlled according to duty cycles. Especially, the flow rate at 30% duty cycle is higher than that of 40% duty cycle, and the abrupt increases of the flow rate found in the 20% duty cycle and 40% duty cycle might induce tissue damage. How many measurements have you done for this experiment in each condition? It is required to mention the results with any possible reasons. If it is due to a less filled solution in the reservoir, the experiment needs to be conducted with a fully loaded reservoir or elimination of bubbles in the reservoir.

Response: We thank the reviewer for this comment. In the supplementary figure, the 20%, 40%, and 50% values come from the same device, whereas the 30% value was from another device of an earlier generation. We have elected to represent flow rate vs. duty cycle for a single device in the figure, to avoid confusions related to individual device calibration. We added repeating flow rate measurements with 10% and 20% duty cycle demonstrating the consistency of the fluidic delivery.

Supplementary Figure 14 (g) Maximum flow rate at duty cycle 10% and 20%.

Comment 6: It is required to specify the dimension of the probe implanted in the brain, such as total thickness, width, and length.

Response: We added information on the dimensions of the probe.

Modifications to the manuscript: We clarified this in the text.

Page 6, line 119: "(Probe total thickness: ~350 μm , Distance...)"

Page 29, line 644: "Using the attachment flag, the probes (Length: 2mm) were stereotactically..."

Comment 7: The authors showed only an area of astrocyte and microglia in the histology data. However, the intensity information is also important to compare the biocompatibility. It is required to add intensities of fluorescence signal according to the distance from edges.

Response: We added the intensity information for glial reactivation in the revised manuscript. The average fluorescent intensity of GFAP does not differ between conditions, while the average fluorescent intensity of IBA1 is significant lower in the probe implanted brain tissue, compared to the cannula (**Supplementary Fig. 19**).

REVIEWER COMMENTS

Reviewer #1 (Remarks to the Author):

The authors have adequately addressed the reviewer comments. I recommend acceptance as is.

Reviewer #2 (Remarks to the Author):

The authors have addressed my previous concerns. The publication is recommended. Great job!

Reviewer #3 (Remarks to the Author):

Thank you for taking the time and effort to address reviewer concerns. The authors have mostly addressed all my concerns and now have provided a much-improved version of the manuscript. There are only some details that the authors should revise:

1. The authors performed new experiments to show the technical advances of the proposed system. They designed two logics of pump/LED operation, and the results showed different behavioral outcomes even though the two logics had the same amount of pump and LED operation time. It is required to discuss any possible reasons.
2. The authors mentioned that the flow rate data in Supplementary Figure 14 was measured from another device. However, it is still unclear that the calculated max flow rate of 30% duty cycle is higher than the max flow rate of 40% duty cycle. It is required to mention the reason why the calculated max flow rate of 30% duty is higher than those of 20% and 40%. If the inconsistency came from different dimensions of the device used in the measurement of the 30% duty cycle, you need to mention it in the manuscript.
3. The flow rates of 20% duty cycle and 40% duty cycle seem to highly fluctuate during operation, which was now shown in 10%, 30%, and 50% duty cycles. Please discuss any possible reasons.
4. In the proposed pump, the amount of delivered solution also seems to be critical. Please discuss whether the amount of solution can be controlled by the duty cycle by adding a plot of the delivered solution vs duty cycles.
5. Reference #42 does not seem to have contents about the agarose gel model. Please check the reference again and replace the reference if needed.

Summary of revisions

We thank the Reviewers for their support and excellent suggestions. We modified the manuscript to address the remaining concerns.

Major changes to the manuscript text are highlighted in red color. Line numbers are also referenced throughout responses to specific comments.

Response to Reviewer 3

Comment: 1. The authors performed new experiments to show the technical advances of the proposed system. They designed two logics of pump/LED operation, and the results showed different behavioral outcomes even though the two logics had the same amount of pump and LED operation time. It is required to discuss any possible reasons.

Response: The different behavioral outcomes between the two logics may be explained by the real-time concentration of Rubi-glutamate in the local brain region. In logic 1, continuous pump operation may result in a high peak concentration that will decay exponentially based on the diffusion kinetics of brain tissue (Wolak and Thorne, 2013). In logic 2, intermittent pump operations will maintain a relatively more stable concentration of Rubi-glutamate around the probe throughout the 6-min period. Thus, intermittent pump operations may allow the researcher to extend the pharmacological effects with a limited volume of drugs. Other dosing regimens can also be easily adapted by optimizing the delivery logic based on the pharmacokinetics of desired compounds.

Modifications to the manuscript:

Page 20, line 438: "The pump operation (e.g., continuous vs intermittent delivery) will result in distinct local drug concentration patterns that may differentially regulate the neuronal activity of the targeted brain region."

Comment: 2. The authors mentioned that the flow rate data in Supplementary Figure 14 was measured from another device. However, it is still unclear that the calculated max flow rate of 30% duty cycle is higher than the max flow rate of 40% duty cycle. It is required to mention the reason why the calculated max flow rate of 30% duty is higher than those of 20% and 40%. If the inconsistency came from different dimensions of the device used in the measurement of the 30% duty cycle, you need to mention it in the manuscript.

Response: We believe that differences between devices cause this apparent inconsistency. To avoid confusion for the readers, we decided to remove the 30% duty cycle data because it was generated using a device with a slightly different design compared to that for the other cases.

Modifications to the manuscript:

Comment: 3. The flow rates of 20% duty cycle and 40% duty cycle seem to highly fluctuate during operation, which was now shown in 10%, 30%, and 50% duty cycles. Please discuss any possible reasons.

Response: These fluctuations represent uncertainties due to insufficient or variable numbers of tracking particles and/or non-uniform distributions of these particles due to aggregation or accumulation at the channel boundaries.

Modifications to the manuscript:

Page 12, line 267: "The measurement uncertainties are likely due to insufficient or variable numbers of tracking particles and/or non-uniform distributions of these particles due to aggregation or accumulation at the channel boundaries."

Comment: 4. In the proposed pump, the amount of delivered solution also seems to be critical. Please discuss whether the amount of solution can be controlled by the duty cycle by adding a plot of the delivered solution vs duty cycles.

Response: **Supplementary Figure 14g** shows the delivered drug volume as a function of time for different duty cycles. The measured values of the effective current were used in the model. The results of this modeling yield delivered volumes as a function of duty cycle at 5 seconds (**Supplementary Figure 14h**). The findings demonstrate that the delivered drug volume can be controlled by duty cycle.

Modifications to the manuscript:

Page 13, line 269: “The results of modeling confirm the capability to control the delivered volume through the duty cycle and delivery time (**Supplementary Figure 14h,i**). Using a simple GUI, the duty cycles, delivery times, delivery cycles and interval times can be easily configured, thereby allowing multi-dose animal experiments.”

Comment: 5. Reference #42 does not seem to have contents about the agarose gel model. Please check the reference again and replace the reference if needed.

Response: We agree. We replaced Reference #42 with “Chen, Z.-J. et al. A realistic brain tissue phantom for intraparenchymal infusion studies. *J. Neurosurg.* 101, 314–322 (2004).”